# Human primed ILCPs support endothelial activation through NF-κB signaling

**Giulia Vanoni[1], Giuseppe Ercolano[1], Simona Candiani[2], Mariangela Rutigliani[3], Mariangela Lanata[3], Laurent Derré[4], Emanuela Marcenaro[5], Pascal Schneider[6], Pedro Romero[7], Camilla Jandus[1]\*, Sara Trabanelli[1†]\***

[1]Department of Oncology, Ludwig Institute for Cancer Research - University of Lausanne, Lausanne, Switzerland; [2]Department of Earth Science, Environment and Life, University of Genova, Genova, Italy; [3]Department of Laboratory and Service, Histological and Anatomical Pathology, E.O. Galliera Hospital, Genova, Italy; [4]Department of Urology, University Hospital of Lausanne (CHUV), Lausanne, Switzerland; [5]Department of Experimental Medicine and Centre of Excellence for Biomedical Research, University of Genova, Genova, Italy; [6]Department of Biochemistry, University of Lausanne, Lausanne, Switzerland; [7]Department of Oncology, University of Lausanne, Lausanne, Switzerland

**\*For correspondence:**
camilla.jandus@unige.ch (CJ);
sara.trabanelli@unige.ch (ST)

**Present address:** [†]Department of Pathology and Immunology, Faculty of Medicine, University of Geneva, Geneva, Switzerland

**Competing interests:** The authors declare that no competing interests exist.

**Abstract** Innate lymphoid cells (ILCs) represent the most recently identified subset of effector lymphocytes, with key roles in the orchestration of early immune responses. Despite their established involvement in the pathogenesis of many inflammatory disorders, the role of ILCs in cancer remains poorly defined. Here we assessed whether human ILCs can actively interact with the endothelium to promote tumor growth control, favoring immune cell adhesion. We show that, among all ILC subsets, ILCPs elicited the strongest upregulation of adhesion molecules in endothelial cells (ECs) in vitro, mainly in a contact-dependent manner through the tumor necrosis factor receptor- and RANK-dependent engagement of the NF-κB pathway. Moreover, the ILCP-mediated activation of the ECs resulted to be functional by fostering the adhesion of other innate and adaptive immune cells. Interestingly, pre-exposure of ILCPs to human tumor cell lines strongly impaired this capacity. Hence, the ILCP–EC interaction might represent an attractive target to regulate the immune cell trafficking to tumor sites and, therefore, the establishment of an anti-tumor immune response.

## Introduction

Innate lymphoid cells (ILCs) constitute the latest described family of innate lymphocytes with key functions in the preservation of epithelial integrity and tissue immunity throughout the body (*Mjösberg and Spits, 2016*). Besides conventional natural killer (cNK) cells, three main distinct subsets of non-NK helper-like ILCs have been described so far, mirroring the transcriptional and functional phenotype of CD4[+] T helper (Th) cell subsets (*Diefenbach et al., 2014*): ILC1s, ILC2s, and ILC3s, that mainly produce IFN-γ, IL-4/IL-5/IL-13, and IL-17A/IL-22 respectively (*Mjösberg and Spits, 2016*).

   In human tissues, the majority of ILCs is mainly terminally differentiated, while a population of circulating Lin[-] CD127[+]CD117[+]CRTH2[-] ILCs, able to differentiate into all ILC subsets, has been recently identified in the periphery and named ILC precursors (ILCPs, *Lim et al., 2017*). ILCPs are characterized by the expression of CD62L that drives their migration to the lymph nodes (*Bar-Ephraim et al., 2019*). Enriched at surface barriers, ILCs rely on IL-7 for their development and

promptly respond to tissue- and cell-derived signals by producing effector cytokines in an antigen-independent manner (*Nussbaum et al., 2017*).

The different ILC subsets have important effector functions during the early stages of the immune response against microbes, in tissue repair and in the anatomical containment of commensals at surface barriers (*Hazenberg and Spits, 2014*). In addition, depending on the ILC subset that is involved and on the tumor type (*Salomé and Jandus, 2018*; *Chiossone et al., 2018*; *Ercolano et al., 2019*; *Ercolano et al., 2020*), ILCs have been shown to also exert pro- and anti-tumoral activity by interacting with different cell types, including endothelial and stromal cells. In a subcutaneous melanoma mouse model, IL-12-responsive NKp46+ ILCs, recruited to the tumor, supported a massive leukocyte infiltration through the upregulation of adhesion molecules in the tumor vasculature (*Eisenring et al., 2010*). In humans, NKp44+ ILC3s were found to be present at early stage in non-small cell lung cancer (NSCLC) patients (*Carrega et al., 2015*) and to correlate with a more favorable prognosis, possibly by promoting intratumoral tertiary lymphoid structure (TLS) formation (*Dieu-Nosjean et al., 2008*).

However, scant data are available about the interaction between human ILCs and the vascular endothelium, which constitutes the physical barrier to be crossed by peripheral blood (PB) immune cells to migrate into tissues where to exert their effector functions (*Nourshargh et al., 2010*).

In this study, we show for the first time that human primed ILCPs can interact with endothelial cells (ECs), upregulate adhesion molecules, and stimulate their pro-inflammatory cytokine secretion. This activation occurs through NF-κB, primarily in a contact-dependent manner that engages surface TNF and RANKL. We report that the ILCP-mediated activation of the ECs is functional, i.e., it allows the adhesion of freshly isolated PB immune cells. Moreover, we show that the ability of ILCPs to activate ECs is dampened after the co-culture with tumor cells. With this study, we have unraveled a cell intrinsic ability of ILCPs that might be selectively impaired by tumors to favor their immune escape.

## Results

### ILCPs upregulate adhesion molecules on EC surface and acquire an activated and ILC3-like phenotype in vitro

The first evidence of an ILC–EC interaction was reported by Eisenring and colleagues in an in vivo melanoma model (*Eisenring et al., 2010*). To investigate whether also human ILCs can interact with ECs, individual circulating ILC subsets, identified based on the expression of c-Kit and CRTH2 within the Lin⁻ CD127+ fraction (*Figure 1a*), were ex vivo-sorted from the PB of healthy volunteers and short-term in vitro-expanded and eventually re-sorted at a purity ≥90%, before use in co-culture experiments with primary human ECs (HUVECs, *Figure 1—figure supplement 1a*). Upon exposure of ECs to in vitro-expanded ILC subsets, ILCPs were the only subset that significantly upregulated the adhesion molecules E-Selectin, ICAM-1, and VCAM-1 on the EC surface, if compared to ILC1s and ILC2s (*Figure 1b*). These adhesion proteins are involved in the different stages of the multi-step process of the leukocyte transendothelial migration (TEM) process, i.e., the movement of leukocytes out of the blood stream and toward the site of tissue damage and/or infection (*Muller, 2011*). We confirmed the ability of in vitro-expanded ILCPs to activate ECs using other primary human dermal blood ECs, i.e., HDBECs (*Figure 1—figure supplement 1b*). Following in vitro expansion, we observed that ILCPs upregulated NKp44 and CD69 as well as CD45RO and RORγt, if compared to their ex vivo counterparts, while maintaining similar levels of expression of NKp46 and CD62L (*Figure 1—figure supplement 1c and d*), suggesting that the in vitro expansion process conferred a more committed phenotype to this ILC subset. Interestingly, no difference in T-bet or GATA3 expression in RORγt+ vs RORγt⁻ cells was observed (*Figure 1—figure supplement 1e*), indicating that the expression or not of RORγt is not directly involved in the EC-activating capacity of ILCPs. Since we observed that around 60% of in vitro-expanded ILCPs acquired NKp44 expression, we investigated the ability of NKp44+ vs NKp44⁻ ILCPs to activate ECs. As shown in *Figure 1—figure supplement 1f*, no significant difference was observed in the EC-activating capacity of these two subpopulations, suggesting that the EC-activating capacity of ILCPs does not depend on the expression of NKp44. Moreover, we observed that in vitro-expanded ILCPs upregulated the expression of the chemokine receptors CCR6 and CXCR5, i.e., two known LTi-like cells markers, compared to their ex vivo counterparts. Consistent with previous reports, Neuropilin1 (NRP1) was not expressed by

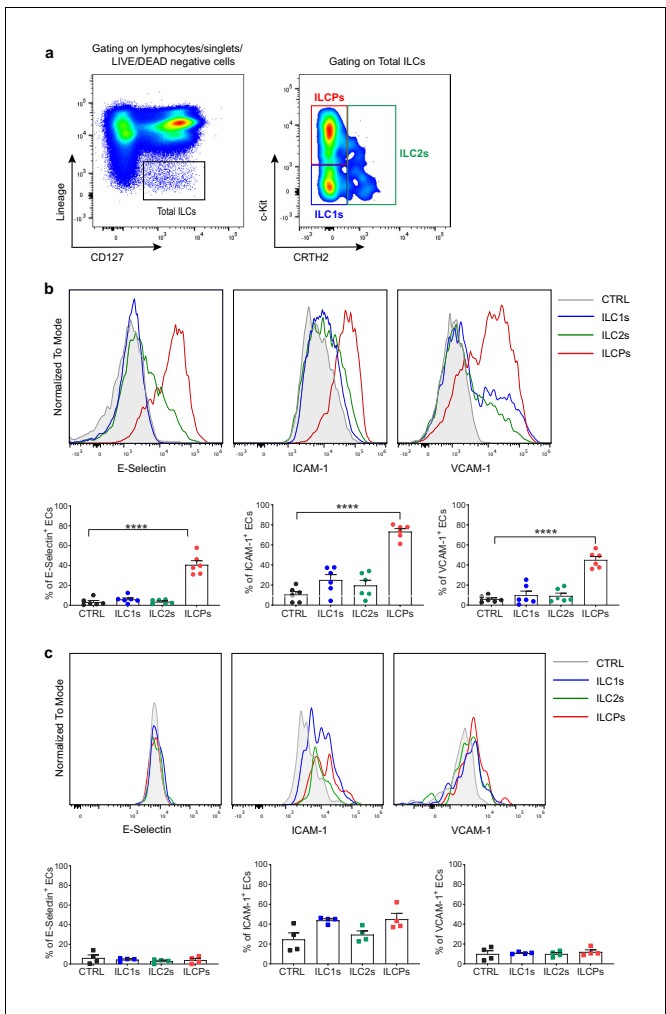

**Figure 1.** In vitro-expanded innate lymphoid cell precursors (ILCPs) induce adhesion molecule expression in endothelial cells (ECs). (a) Circulating human ILCs are identified as lineage negative CD127$^+$ cells; within this population, we discriminate ILC1s as c-Kit$^-$ CRTH2$^-$, ILC2s as CRTH2$^+$ c-Kit$^{+/-}$, and ILCPs as c-Kit$^+$ CRTH2$^-$ cells. HUVEC cells were co-cultured for 3 hr at 1:1 ratio in direct contact with either in vitro-expanded (b) or directly ex vivo-sorted (c) ILC1s, ILC2s, and ILCPs. Untreated ECs were employed as negative control (CTRL). ECs were harvested and analyzed for cell-surface adhesion molecule expression by flow cytometry. Graphs show representative histograms (panels b and c, top) and the summary (panels b and c, bottom) of the induction of the indicated adhesion molecules on the EC surface (n = 6). Ordinary one-way ANOVA–Tukey's multiple comparison test (panel b); Ordinary one-way ANOVA–Friedman test (panel c).

The online version of this article includes the following source data and figure supplement(s) for figure 1:

**Source data 1.** Raw data of panels b and c.

**Figure supplement 1.** Human innate lymphoid cell precursors (ILCPs) acquire an activated phenotype in vitro.

**Figure supplement 1—source data 1.** Raw data of panels b and d–f.

**Figure supplement 2.** In vitro-expanded Th subsets fail to activate endothelial cells (ECs).

**Figure supplement 2—source data 1.** Raw data of panel b.

circulating ILC3s (*Shikhagaie et al., 2017*) and was not upregulated after in vitro expansion. Compared to ex vivo ILCPs, in vitro-expanded ILCPs downregulated the expression of CD28, although only 20% of circulating ILCPs expressed it (*Figure 1—figure supplement 1c and d*). Overall, these data suggest that not only in vitro-expanded ILCPs acquire an activated phenotype in vitro, but are also skewed toward an ILC3-like phenotype and share some phenotypical markers with LTi-like cells, while maintaining multipotent features as shown by the expression of T-bet and GATA3.

To understand if the ability of ILCPs to interact with ECs is an intrinsic property of these cells or if they need to be primed to acquire it, we decided to expose ECs directly to ex vivo-sorted ILC subsets. As shown in *Figure 1c*, none of the isolated ILC subsets could induce a significant activation of ECs, suggesting that the EC-activating capacity of ILCPs is acquired during the in vitro expansion process. Since ILCPs were expanded in the presence of feeder cells, PHA, and IL-2, it is conceivable that feeder-derived cytokines such as IL-12 and IL-1β are involved in the priming. As ILCs constitute the innate counterpart of CD4$^+$ T cells, we tested if in vitro-expanded individual T-helper (Th) subsets, i.e., Th1, Th2, Th17, and Th* (i.e., Th cells with a Th1/Th17 intermediate phenotype [*Sallusto, 2016*; *Figure 1—figure supplement 2a*]) could also interact, at steady state, with ECs. Following the same expansion protocol employed for ex vivo-isolated ILC subsets, Th subsets were employed in 3 hr co-culture experiments with ECs. As reported in the *Figure 1—figure supplement 2b*, except for a statistically significant Th1-mediated upregulation of VCAM-1, still not to the same extent as the ILCP-mediated induction, all Th subsets failed to upregulate adhesion molecule expression on the EC surface. Overall, these data suggest that in vitro-expanded ILCPs not only acquire a more activated/ILC3-like phenotype in vitro, but also the ability of interacting with ECs by means of mediating the upregulation of adhesion molecule expression on the EC surface.

## ILCPs activate ECs primarily in a contact-dependent mechanism

Inflammation triggers the upregulation of adhesion molecules in ECs, promoting the accumulation of leukocytes and their adhesion to the blood vessel walls. This phenomenon is mediated by pro-inflammatory mediators, such as TNF and IL-1β (*Collins et al., 1995*). As a consequence, to discriminate whether the EC activation by ILCPs was due to contact-dependent or soluble factor(s)-dependent mechanism(s), supernatants from the EC/ILCP co-cultures were analyzed. Significantly higher levels of IL-6, IL-8, GM-CSF, and IFN-γ were observed (*Figure 2a*). To address which cell type was producing the pro-inflammatory cytokines that accumulate in the cell-free supernatants, qPCR analysis of ECs and ILCPs (CD31-based FACS-sorted after 3 hr co-culture) was performed and compared to untreated ECs and steady-state ILCPs. As reported in *Figure 2b*, high levels of IL-6 and IL-8 transcripts were found in ECs exposed to ILCPs, whereas TNF transcripts were high only in steady-state ILCPs, indicating that IL-6 and IL-8 measured in the supernatant (*Figure 2a*) derive from ECs, and TNF from ILCPs. GM-CSF and IFN-γ transcripts were observed in both ECs and ILCPs before and after co-culture, indicating that both cell types contribute to the accumulation of these two cytokines in the supernatant. To experimentally verify if the upregulation of adhesion molecules in ECs was dependent on these soluble factors, 0.4 µm pore transwell chambers were employed, to allow cytokine exchange between the two compartments yet avoiding the cell contact. In this context, ILCPs failed to induce the expression of adhesion molecules on EC surface (*Figure 2c*). Of note, the production of the pro-inflammatory cytokines was dramatically reduced in the presence of the transwell insert (*Figure 2—figure supplement 1a*). To further prove the direct contact-dependency of the EC–ILC interaction, ECs were incubated during 3 hr in the presence of cell-free supernatant collected from previous EC–ILCP co-culture. As reported in *Figure 2d*, cell-free supernatant did not lead to the upregulation of the adhesion molecules E-Selectin and VCAM-1 in ECs, although ICAM-1 levels were found to be significantly increased if compared to unstimulated ECs, yet not to the same extent as for ILCP-exposed ECs. Finally, we analyzed the production of IL-6, IL-8, TNF, GM-CSF, and IFN-γ by ex vivo- and in vitro-expanded ILCPs. As shown in *Figure 2—figure supplement 1b*, no difference in terms of secretion of the indicated cytokines was observed. Indeed, incubation of ECs during 3 hr with cell-free supernatant collected from pure ILCPs at the end of the in vitro expansion did not provoke upregulation of adhesion molecules on EC surface (*Figure 2—figure supplement 1c*), correlating with the very low amount of the pro-inflammatory cytokines as shown in *Figure 2—figure supplement 1b*. Overall, these data suggest that ILCPs are superior to other ILC subsets in inducing the upregulation of adhesion molecules on ECs and can also favor the release of pro-inflammatory cytokines, primarily in a contact-dependent manner.

## ILCPs engage the NF-κB pathway in ECs

It has been shown that adhesion molecule expression can be induced in ECs during inflammatory responses by the activation of different signaling pathways, among which the NF-κB pathway (*Rahman and Fazal, 2011*). To test whether the induction of adhesion molecules by ILCPs was

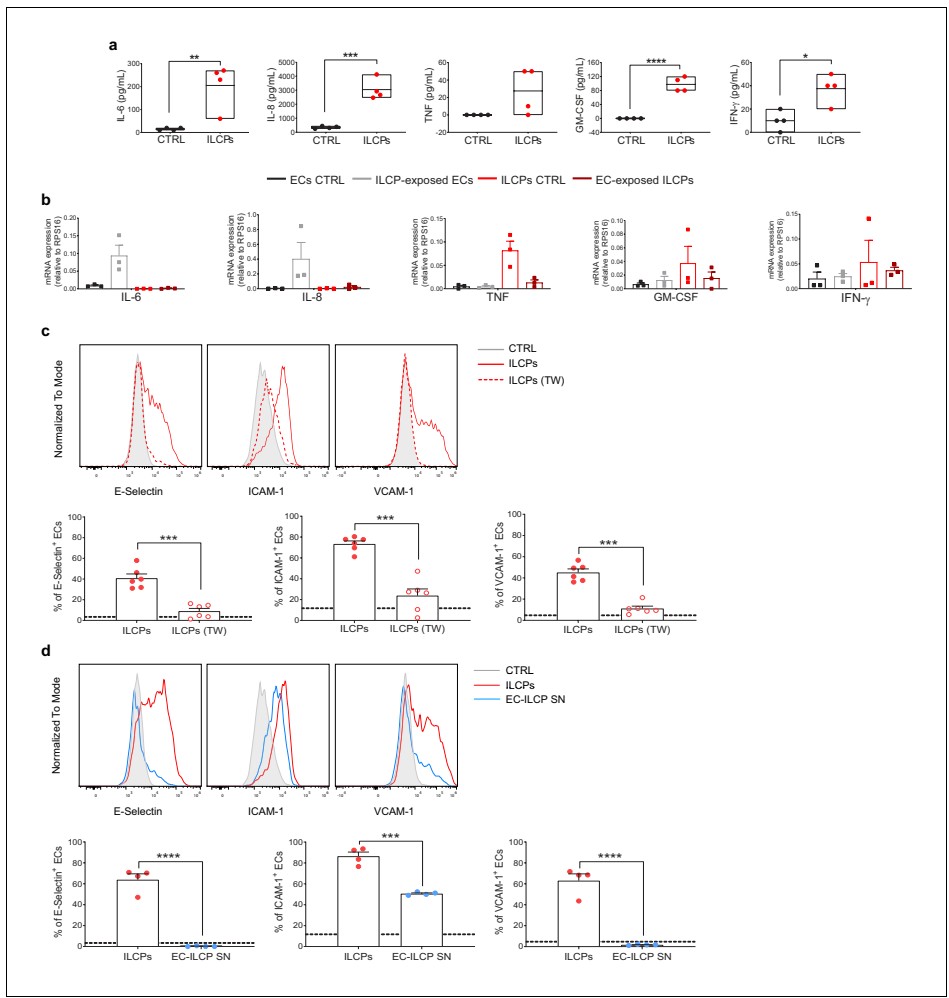

**Figure 2.** Human innate lymphoid cell precursors (ILCPs) activate ECs primarily in a contact-dependent mechanism in vitro. (**a**) The supernatant of the 3 hr co-culture experiments between ECs and ILCPs was analyzed for its cytokine contents (n = 4). The composition of the supernatant of ECs in EC growth medium was used as negative control (CTRL). (**b**) The expression of IL-6, IL-8, GM-CSF, TNF, and IFN-γ was analyzed by qPCR in ECs and ILCPs after being cultured for 3 hr at 1:1 ratio and FACS-sorted according to CD31 expression. Untreated ECs and ILCPs were employed as controls (CTRL). (**c**) HUVEC cells were co-cultured for 3 hr at 1:1 ratio in direct contact with in vitro-expanded ILCPs either in the absence (red dots) or presence (red circles) of a transwell (TW) insert (0.4 μm pore polycarbonate filter) or (**d**) in the presence of pre-conditioned media coming from previous EC–ILCP 3 hr co-cultures. ECs were harvested and analyzed for cell-surface adhesion molecule expression by flow cytometry (n = 6). The dotted lines indicate the level of average expression of adhesion molecules by unstimulated ECs. Statistical tests used: Unpaired t-test (panels **a** and **d**); paired t-test (panel **c**).

The online version of this article includes the following source data and figure supplement(s) for figure 2:

**Source data 1.** Raw data of panels a–d.

**Figure supplement 1.** Human innate lymphoid cell precursors (ILCPs) activate endothelial cells (ECs) primarily in a contact-dependent manner in vitro.

**Figure supplement 1—source data 1.** Raw data of panels a–d.

---

dependent on NF-κB, ECs were pre-treated during 1 hr with a IκB kinase (IKK) complex inhibitor (BAY 11–7082, *Mori et al., 2002*) to specifically prevent NF-κB activation. In this context, ILCPs failed to significantly induce the expression of adhesion molecules on pre-treated ECs (*Figure 3a*), indicating that ILCPs need to engage the NF-κB pathway to activate ECs in vitro. Similar to what we observed in the context of ILCPs cultured with ECs in the presence of a transwell insert, the prevention of NF-κB activation in ECs led to a significant decrease of IL-6, as well as reduction in IL-8, GM-CSF, and IFN-γ secretion (*Figure 2—figure supplement 1d*). Next, to understand which molecular

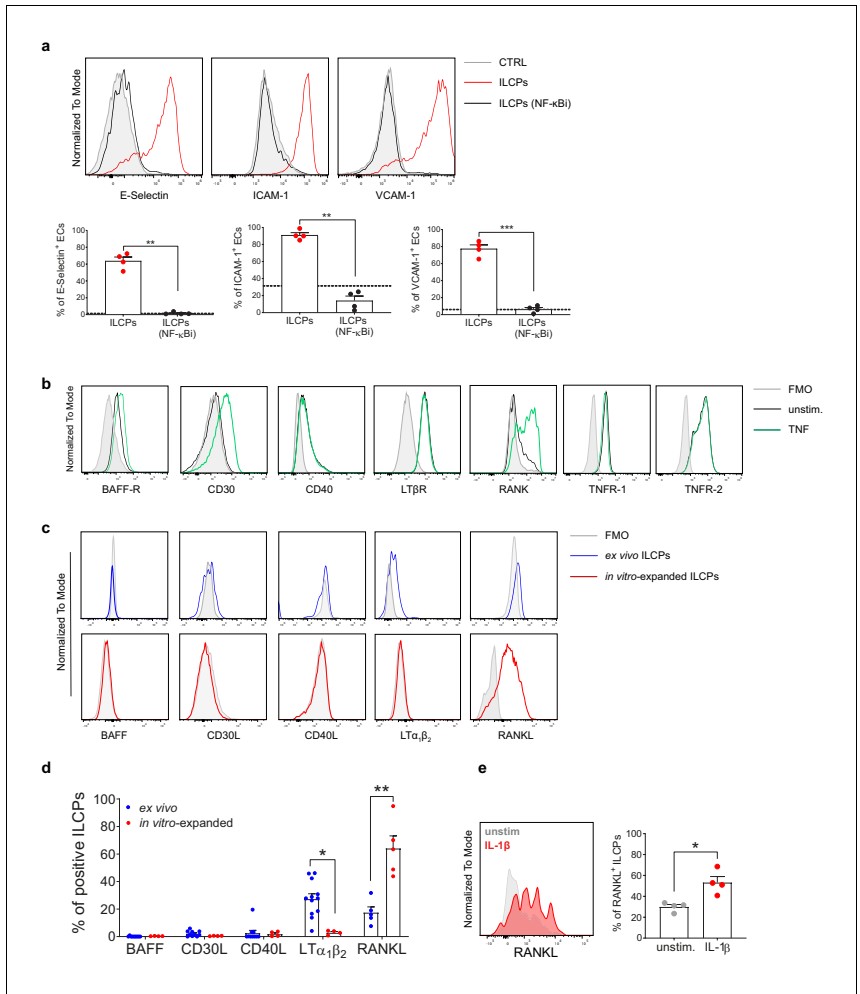

**Figure 3.** Innate lymphoid cell precursors (ILCPs) induce adhesion molecules expression on the endothelial cell (EC) surface via NF-κB pathway activation. (**a**) HUVEC cells were treated during 1 hr with 2.5 μM of a specific inhibitor of both canonical and alternative NF-κB pathways (BAY 11–7082, Adipogen) and then exposed to ILCPs at 1:1 ratio for 3 hr. ECs were harvested and analyzed for cell-surface adhesion molecule expression by flow cytometry (n = 4). The black dotted line indicates the level of average expression by untreated ECs. (**b**) HUVEC cells were tested for the expression of NF-κB activating receptors, either at steady-state (black line) or following 3 hr in vitro stimulation with 20 ng/mL of TNF (green line). (**c and d**) The respective activating ligands were analyzed on both ex vivo- and in vitro-expanded ILCPs. Graphs show representative histograms (panel **c**) and the summary (panel **d**) of the analysis performed on HDs (n = 4–11). (**e**) In vitro-expanded ILCPs were stimulated during 24 hr in the presence of 20 ng/mL of IL-1β or left untreated and stained for surface RANKL (n = 4). Statistical tests used: Paired t-test (panels **a and e**); Multiple t-tests (panel **d**).

The online version of this article includes the following source data for figure 3:

**Source data 1.** Raw data of panels a, d, and e.

players were involved in the ILC–EC cross-talk, we screened ECs and, both ex vivo and in vitro-expanded, ILCPs for the presence on their surface of receptors and ligands, respectively, known to be involved in the NF-κB pathway activation. On one side, we observed that untreated ECs constitutively expressed the lymphotoxin-β receptor (LT-βR), as well as the tumor necrosis factor (TNF) receptors 1 and 2 (TNFR-1 and TNFR-2, respectively), whereas B-cell activating factor receptor (BAFF-R), CD40, and RANK were expressed only at low levels (*Figure 3b*). Following stimulation with TNF, CD30 expression became detectable and BAFF-R and RANK expression increased, while CD40 and LT-βR expression remained unchanged (*Figure 3b*). On the other side, when looking at extracellular NF-κB activating ligands on ex vivo ILCPs, we observed that they expressed high levels

of the transmembrane form of lymphotoxin (LTα₁β₂), a described ligand for LT-βR (*Madge et al., 2008*), if compared to in vitro-expanded ILCPs (*Figure 3c and d*). Both BAFF and CD30L were undetectable and low levels of CD40L and RANKL were observed. In contrast, in vitro-expanded ILCPs upregulated the expression of RANKL and downregulated that of LTα₁β₂ (*Figure 3c and d*). It has been reported that pro-inflammatory cytokines, such as interleukin 12 (IL-12), can induce RANKL on human periodontal ligament cells in vitro (*Issaranggun Na Ayuthaya et al., 2017*). Since it is known that feeder cells can produce a wide array of cytokines, among which IL-1β and IL-12, we decided to test whether RANKL expression might be upregulated by one of these factors. Surprisingly, after 24 hr stimulation of freshly ex vivo isolated ILCPs with IL-1β (*Figure 3e*), but not with IL-12 (data not shown), we observed increased expression of RANKL compared to untreated ILCPs. The transmembrane form of TNF (tm-TNF) constitutes another described NF-κB activating ligand. However, the detection of the membrane-bound form of TNF could not be tested due to the lack of a specific antibody. Moreover, the discrimination between the soluble and the membrane forms of TNF at mRNA levels is not possible, since TNF is transcribed (and also translated) as a full-length membrane-bound precursor (*Black et al., 1997*). However, at the end of the in vitro expansion, ILCPs showed higher levels of TNF transcripts compared to ex vivo ILCPs (data not shown). Overall, these data show that in vitro-expanded ILCPs express TNF, possibly present on the ILCP surface, to in vitro interact with ECs via TNFRs and upregulate RANKL expression, possibly via feeder-cell-derived IL-1β, to engage RANK on ECs.

## ILCPs activate ECs via the engagement of TNFR and RANK

To test which of the NF-κB activating molecules was responsible for the upregulation of adhesion molecules on EC surface, a series of blocking experiments using different soluble Fc fusion proteins were performed to prevent the binding of defined ligands to their receptors on ILCPs. Since we observed increased levels of RANKL on in vitro-expanded ILCPs as compared to their ex vivo counterparts (*Figure 3e*), and higher levels or RANK on ECs following 3 hr co-culture with ILCPs (*Figure 4—figure supplement 1a*), we decided to interfere with the RANK/RANKL interaction. As negative control, we performed the blocking experiments with intravenous immune globulins (IVIGs), a pool of human gamma globulins (*Figure 4—figure supplement 1b*). Although ILCPs were still able to activate ECs in this setting with yet an inhibition of E-Selectin triggering in ECs (*Figure 4a*), we observed that the levels of IL-6 and GM-CSF were dramatically reduced, if compared to the cytokine composition of ECs cultured with steady-state ILCPs (*Figure 4—figure supplement 1c*). Therefore, we hypothesized a major involvement of tm-TNF in the induction of adhesion molecules. Thus, we pre-incubated ILCPs in the presence of TNFR1:Fc and/or TNFR2:Fc and we observed that the EC expression of adhesion molecules was significantly reduced (*Figure 4b*). In all cases, inhibition with TNFR2:Fc was slightly more efficient than with TNFR1:Fc, which could be explained by the greater affinity of TNFR2 for TNF (*Grell et al., 1995*). Of note, no difference in the cytokine secreted levels was observed (*Figure 4—figure supplement 1d*), suggesting that interfering with the TNF-TNFR signaling does not impact cytokine production in both cell types. Addition of RANK:Fc to TNFR1:Fc and TNFR2:Fc further slightly reduced E-Selectin, ICAM-1 and VCAM-1 levels, although the contribution of RANK:Fc was not significant (*Figure 4c*). However, we could observe a decreased production of the pro-inflammatory cytokine IL-6, IL-8, TNF, and GM-CSF (*Figure 4—figure supplement 1e*) when blocking ligands of TNFR1, TNFR2, and RANK in ILCPs/ECs co-cultures. Taken together, our data suggest that ILCPs activate EC primarily through the engagement of TNFRs to upregulate adhesion molecules expression on EC surface. The engagement of RANK in ECs does not seem to have an additive effect in inducing adhesion molecules expression, but might act in synergy with tm-TNF to control the cytokine secretion and further support the EC activation.

## ILCP-mediated EC activation favors the adhesion of freshly isolated PBMCs in vitro

To address the functionality of the EC–ILCP interaction, i.e., the adhesion of freshly isolated PBMCs to ILCP-exposed EC, a static adhesion assay was performed. Briefly, following the 3 hr co-cultures, CD31⁺ ECs were isolated by FACS, to remove adherent ILCPs, and re-plated. After the sorting, untreated ECs (negative control) did not upregulate adhesion molecule expression on their cell surface, and ILCP-exposed ECs maintained comparable surface levels of adhesion molecule as before

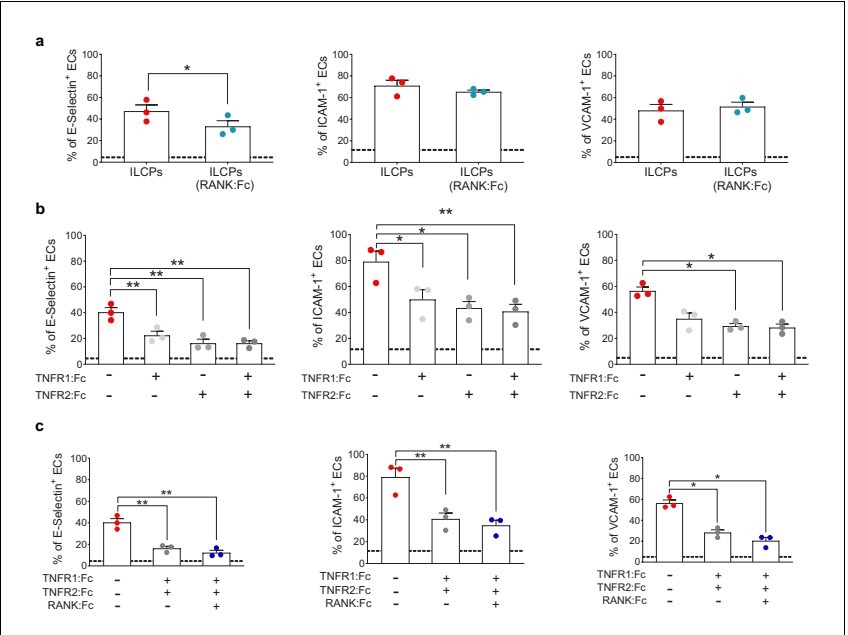

**Figure 4.** Innate lymphoid cell precursor (ILCP)-mediated upregulation of adhesion molecules on ECs involves the engagement of TNFR1, TNFR2, and RANK. ILCPs were incubated overnight in the presence of 10 U/mL of rhIL-2 and an additional pre-incubation of 30 min (prior co-culture with ECs) was performed in the presence of 5 μg/mL of RANK:Fc (a), of 2 μg/mL of TNFR1:Fc, 5 μg/mL of TNFR2:Fc, and 5 μg/mL of RANK:Fc, either alone or in combination (b and c). ECs were harvested and analyzed for cell-surface adhesion molecule expression by flow cytometry (n = 3). The dotted lines indicate the level of average expression of adhesion molecules by unstimulated ECs. Statistical test used: Paired t-test.

The online version of this article includes the following source data and figure supplement(s) for figure 4:

**Source data 1.** Raw data of panels a–c.

**Figure supplement 1.** Innate lymphoid cell precursor (ILCP)-mediated modulation of RANK expression on endothelial cell (EC) surface.

**Figure supplement 1—source data 1.** Raw data of panels a–d.

the FACS isolation procedure, showing that the sorting procedure did not affect the activation state of ECs in any of the conditions (*Figure 5a*). The day after, the assay was performed and ECs, together with adherent PBMCs, were detached and stained for flow cytometry analyses. Interestingly, ECs pre-exposed to ILCPs led to the adhesion of a significantly higher number of freshly isolated PBMCs compared to unstimulated ECs. As shown in *Figure 5b and c*, the ILCP modification of EC allowed a strong adhesion of T, B as well as NK cells and monocytes. To understand if the adhesion of freshly isolated PBMCs is itself dependent on NF-kB, we repeated the experiment by exposing untreated or NF-κB-inhibited ECs to TNF for 3 hr the day before performing the static adhesion assay. As shown in *Figure 5d*, the inhibition of NF-κB activation prior stimulation with TNF strongly reduced the numbers of adhered T, B, NK cells, and monocytes. In this setting, we could also observe that ILCs themselves could adhere to TNF-treated ECs (*Figure 5d*). Interestingly, a trend for a reduction in the number of adhered PBMCs to ECs was also observed when NF-κB activation was prevented in ECs 30 min before performing the static adhesion assay (*Figure 5d*) although not significant. Since we showed that NF-κB engagement is crucial for the ILCP-mediated adhesion molecule upregulation in ECs (*Figure 3a*), it was not surprising to observe the impaired adhesion of PBMCs to ECs in vitro. Overall, these data suggest that the adhesion molecule expression induced by the ILCPs is functional, i.e., it supports the adhesion of other immune cell types to ECs in vitro and relies on NF-κB activation.

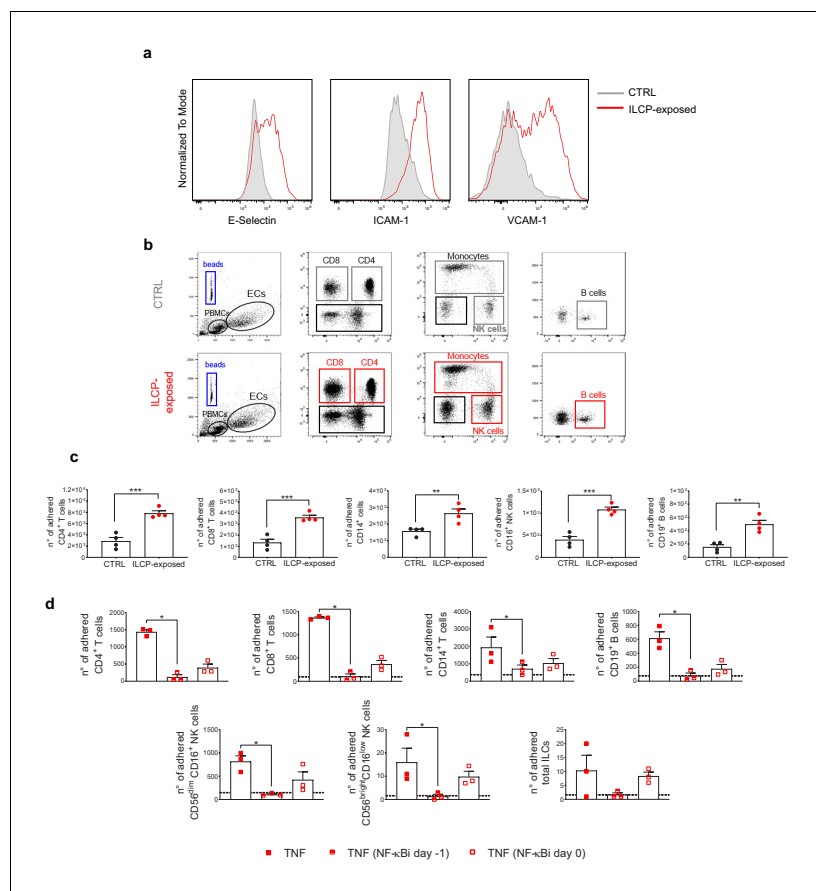

**Figure 5.** Innate lymphoid cell precursor (ILCP)-exposed endothelial cells (ECs) favor the adhesion of freshly isolated PBMCs in vitro. HUVEC cells were co-cultured for 3 hr at 1:1 ratio in direct contact with in vitro-expanded ILCPs or left untreated (CTRL). ECs were harvested, FACS isolated to remove adhered ILCPs, and re-seeded. (a) The graphs show the level of expression of adhesion molecules by ILCP-exposed ECs after the sorting and before performing the static adhesion assay, compared to untreated ECs (gray). Graphs show representative dot plots (b) and the summary (c) of the number of adhered CD3, CD4, CD8, CD14, CD16, and CD19 expressing cells assessed by flow cytometry with the use of CountBright Absolute Counting Beads (blue gate in the dot plots). (d) The day before the assay, HUVEC cells were cultured for 3 hr in the presence of 20 ng/mL of TNF and treated during 1 hr with 2.5 μM NF-κB inhibitor BAY 11–7082 (Adipogen), either before the TNF treatment (half-full red square dots) or directly on the day of the assay (empty red square dots), before incubation with total PBMCs at 1:4 ratio for 30 min. The graphs show the summary of the number of adhered CD3, CD4, CD8, CD14,, CD56$^{dim}$CD16$^{+}$, CD56$^{bright}$CD16$^{low}$, CD19 expressing cells, and ILCs assessed by flow cytometry with the use of CountBright Absolute Counting Beads. Statistical test used: Unpaired t-test (panels **c** and **d**).

The online version of this article includes the following source data for figure 5:

**Source data 1.** Raw data of panels c and d.

## Tumor-derived factors impair ILCP ability to activate ECs in vitro

The poorly functional and altered structural organization of the vascular bed has an important impact on tumor progression and affects endothelial–leukocyte interactions (*Cedervall et al., 2015*). Hence, we were interested in studying the impact that the tumor and/or the tumor microenvironment could exert on ILCPs and, therefore, on their ability to modulate the EC activation. First, we observed that CD3$^{-}$RORγt$^{+}$ ILCs are present in low-grade transitional bladder carcinoma in close proximity to CD31$^{+}$ blood vessels (*Figure 6a*, panels 1–4) but are barely detected in high-grade bladder carcinoma (*Figure 6a*, panels 5–8), suggesting a protective role of RORγt-expressing ILCs, at least at early stage of disease. Interestingly, since we also observed that ILCPs are expanded in the PB of non-muscle-invasive bladder cancer (NMIBC) patients, but reduced in muscle-invasive stage of the

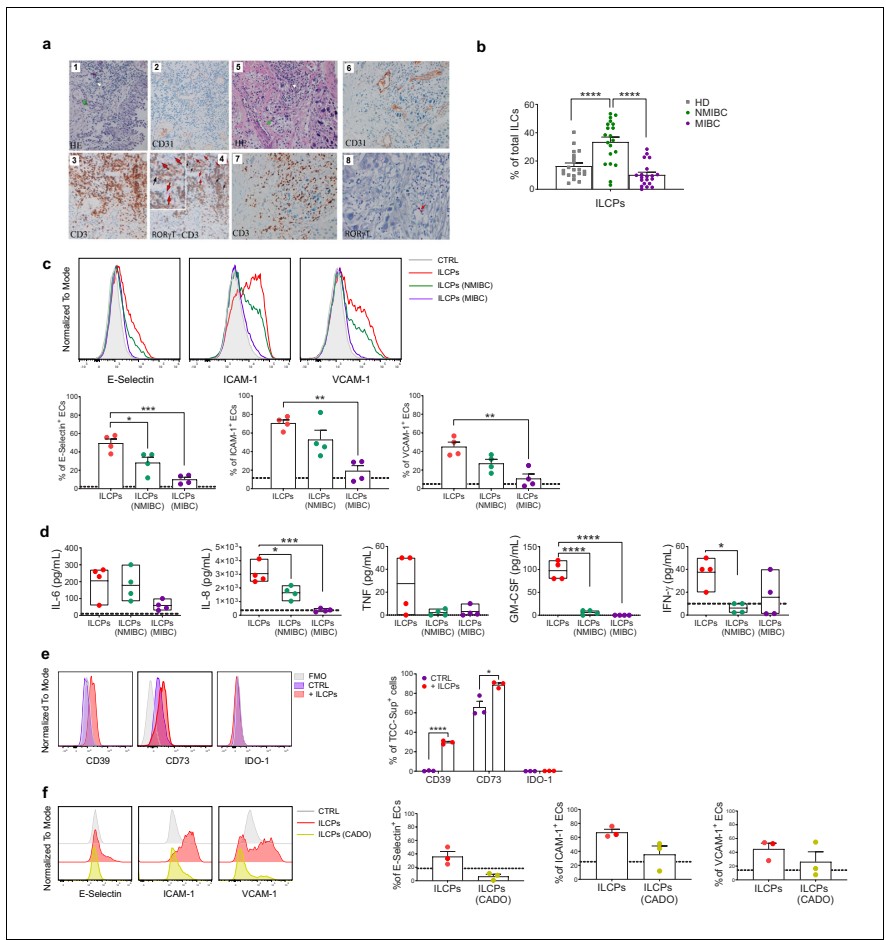

**Figure 6.** Innate lymphoid cell precursors (ILCPs) are found in proximity of blood vessels in low-grade, but barely detected in high-grade, bladder cancer tumor samples and are functionally impaired by co-cultures with bladder carcinoma cells. (a, Panels 1–4) Low-grade transitional bladder cell carcinoma. (a, Panel 1) In the subepithelial connective, blood vessels (green arrow) and inflammatory lymphocytic infiltrate are observed (white arrow) (hematoxylin-eosin staining, 20× magnification). (a, Panel 2) Immunohistochemical CD31 signal showing intense positive endothelial cells (ECs) of blood vessels (20× magnification). (Panel 3) Immunohistochemical detection of CD3[+] cells at level of inflammatory lymphocytic infiltrate (brown signal) (20× magnification). (a, Panel 4) Combined staining with antibody to RORγt and CD3. Black arrows indicate RORγt[+]/CD3[+] cells; red arrows indicate RORγt[+]/CD3[-] cells (red signal) (20× magnification). On the upper left a magnified insert of the main image. Data are representative of five independent experiments. (a, Panels 5–8) High-grade bladder cell carcinoma. (a, Panel 5) In the subepithelial connective, blood vessels (green arrow) and inflammatory lymphocytic infiltrate are observed (white arrow) (hematoxylin-eosin staining, 20× magnification). (a, Panel 6) Immunohistochemical CD31 signal showing intense positive ECs of blood vessels (20× magnification). (a, Panel 7) Immunohistochemical detection of CD3[+] cells at level of inflammatory lymphocytic infiltrate (brown signal) (20× magnification). (a, Panel 8) Immunohistochemical detection of RORγt[+] cells at level of inflammatory lymphocytic infiltrate (red signal, red arrow) (20× magnification). Data are representative of three independent experiments. (b) Flow cytometry characterization of ILCP distribution in the PB of NMIBC and MIBC patients, compared to HDs, expressed as percentage of total ILCs (n = 20). (c) Graphs show representative histograms (panel c, top) and the summary (panel c, bottom) of the induction of adhesion molecules by ILCPs upon different culture conditions, represented as percentage of ECs expressing the indicated adhesion molecules. The dotted lines represent the level of expression of the adhesion molecules in untreated ECs (n = 4). (d) The supernatants of the 3 hr co-culture experiments between ECs and ILCPs, pre-incubated or not for an overnight with bladder carcinoma cell lines, were analyzed for cytokine content (n = 4). The dotted lines indicate the average level of cytokines produced by unstimulated ECs. (e) The expression of CD39, CD73, and IDO-1 in MIBC cells (TCC-Sup) after overnight co-culture with in vitro-expanded ILCPs was assessed by flow cytometry. Untreated TCC-Sup cells (purple bar) were used as controls (CTRL) (n = 3). (f) Graphs show representative histograms (panel f, left) and the summary (panel f, right) of the induction of adhesion molecules by ILCPs pre-treated with 50 µM of 2-Chloroadenosine (a stabilized

*Figure 6 continued on next page*

*Figure 6 continued*

form of adenosine), represented as percentage of ECs expressing the indicated adhesion molecules. The dotted lines represent the level of expression of the adhesion molecules in untreated ECs (n = 3). Statistical tests used: Ordinary one-way ANOVA, Tukey's multiple comparison tests (panels b, c, and d); Multiple t-tests (panel f).
The online version of this article includes the following source data and figure supplement(s) for figure 6:

**Source data 1.** Raw data of panels b–f.
**Figure supplement 1.** Innate lymphoid cell precursors (ILCPs) are found in proximity of blood vessels in low-grade colon adenocarcinoma and are impaired by co-cultures with colon adenocarcinoma cells.
**Figure supplement 1—source data 1.** Raw data of panels b–d.

disease (MIBC, *Figure 6b*), and, following in vitro expansion, ILCPs acquire RORγt expression, we hypothesized that the presence of ILCPs in NMIBC patients might underline the attempt of this cell population to support the infiltration of immune cells into the tumor site. To this aim, ILCPs were pre-exposed to human bladder cancer cell lines, originating either from non-muscle-invasive (NMIBC, early stage) or muscle-invasive (MIBC, late-stage) epithelial bladder carcinoma, thus allowing us to mimic in vitro early and late tumor stage conditions. As shown in *Figure 6c*, the capacity to upregulate adhesion molecule expression on ECs by ILCPs was significantly reduced after the overnight incubation with bladder carcinoma cell lines, if compared to resting ILCPs. Interestingly, the co-culture with MIBC lines showed the highest capacity to modify ILCP ability to activate the ECs (*Figure 6c*). Moreover, the analysis of the cytokine composition of the supernatants from 3 hr EC–ILCP co-culture revealed statistically significant reduced levels of the pro-inflammatory cytokines IL-8, GM-CSF, and IFN-γ when ECs where co-cultured with MIBC pre-exposed ILCPs (*Figure 6d*). Similar observations were obtained using tissue sections of colon adenocarcinoma patients and the SW1116 colon cancer cell line (*Figure 6—figure supplement 1a and b*). To further understand which could be the mechanisms underlying the tumor cell-mediated impairment of ILCPs, we wondered whether the tumor cells were affecting ILCPs via adenosine and/or kynurenines, two metabolites with potent immune-inhibitory effects in the TME (*Vigano et al., 2019*; *Labadie et al., 2019*). As shown in *Figure 6e*, TCC-Sup did not express IDO-1, suggesting that the tumor-mediated effects on ILCPs might not depend on kynurenines. However, following the overnight incubation with ILCPs, TCC-Sup strongly upregulated CD39 and further increased the CD73 expression (*Figure 6e*). Interestingly, steady-state ILCPs also expressed CD39, but very low levels of CD73 (*Figure 6—figure supplement 1c*), suggesting that, in the presence of CD73$^+$ cells, ILCPs might process ATP and support adenosine production. Interestingly, as shown in *Figure 6—figure supplement 1d*, in vitro-expanded ILCPs upregulated the expression of A2A, A2B, and A3 receptors. Of note, pre-exposure of ILCPs to 2-Chloroadenosine (a stabilized form of adenosine) reduced their EC-activating capacity (*Figure 6f*). Taken together, these results suggest that tumor cells might impair or deviate, at least in part via adenosine production, the capacity of ILCs to modulate vascular activation through the upregulation of cell surface adhesion molecules, and affect the production of pro-inflammatory cytokines upon EC–ILCP encounter. Therefore, this could represent a mechanism through which tumors can prevent and block immune cell infiltration into the tumor site.

## Discussion

In this study, we characterized for the first time the in vitro interaction between circulating human ILCs and vascular ECs. In particular we identify ILCPs as the only competent circulating ILC subset in inducing EC activation through the upregulation of adhesion molecules on the EC surface. Our results are consistent with previously reported data showing that group 3 ILCs (defined as Lin$^-$CD127$^+$NKp44$^+$ cells) induce the expression of ICAM-1 and VCAM-1 on mesenchymal stem cells (MSCs) after 4-day co-culture and in the presence of IL-7 (*Cupedo et al., 2009*; *Crellin et al., 2010*). According to recent findings, circulating ILCPs constitute a distinct subset from ILC3s, although they share the expression of c-Kit on their cell surface and are CRTH2$^-$ (*Lim et al., 2017*). Following in vitro priming, the upregulation of RORγt and the expression of activation markers argue for a conversion of ILCPs into committed ILC3-like cells, possibly supported by IL-1β and/or other factors secreted by feeder cells during the expansion phase. This environment mimics the in vivo dynamics observed during inflammatory processes driven by PAMP/DAMP/tumor-dependent DC activation.

However, differently from what was described by Lim and colleagues (*Lim et al., 2017*), the in vitro culture of ILCPs isolated from the PB of HDs did not lead to the expansion of neither ILC1s nor ILC2s, whereas only ILC3-like cells arose. Indeed, the in vitro stimulation applied in that context differs from our in vitro expansion protocol, with the lack of ILC1-, ILC2-, or ILC3-specific cytokines. Overall, our findings support the idea that the in vitro expansion of circulating ILCPs in the presence of feeder cells, PHA, and IL-2 favors their commitment toward an ILC3-like phenotype.

Interestingly, as far as adaptive immune cells are concerned, a previous publication showed that freshly isolated CD4+ CD45RO+ lymphocytes are able to induce, to different extents, the expression of VCAM-1 on ECs in a contact-dependent manner (*Yarwood et al., 2000*). We were unable to recapitulate these findings, most probably due to different culture conditions (isolation and in vitro expansion of T cells, timing, and EC:T-cell ratios). For the innate counterpart, it was shown that the human NK cell line NK92 induces the expression of E-selectin and IL-8 in ECs, which results in EC activation, through the LT-dependent activation of the NF-κB pathway (*von Albertini et al., 1998*). Yet, in our system, we did not observe the upregulation of adhesion molecules when employing in vitro-expanded purified primary NK cells (data not shown).

In this work, we define the ILCP-mediated activation of ECs primarily as a contact-dependent mechanism. However, we cannot exclude that pro-inflammatory cytokines (IL-6, IL-8, GM-CSF, IFN-γ, and TNF), produced during the EC–ILCP interaction, might contribute to the observed EC activation. It is known that pro-inflammatory cytokines, and especially TNF, constitute potent inducers of adhesion molecule expression in ECs (*Collins et al., 1995*). However, in our hands, ECs upregulate adhesion molecules expression only when short-term exposed to TNF, but not to the other cytokines (data not shown). Moreover, the exposure of ECs to cell-free supernatant recovered from foregoing EC–ILCP co-culture only provoked a significant upregulation of ICAM-1 on ECs, and exposure of ECs to cell-free supernatant collected at the end of the in vitro expansion of ILCPs did not upregulate adhesion molecules on ECs, supporting the idea of a primarily contact-mediated interaction between these two cell types. However, it cannot be excluded that the pro-inflammatory cytokines that are produced during the co-culture can support, at the cell–cell contact region, the in vitro cross-talk.

Upon interaction, ILCPs engage the NF-κB pathway in ECs, most probably via TNFR/tm-TNF and RANK/RANKL interactions that possibly act in synergy. RANKL has been recently described as a negative regulator of CCR6+ ILC3s activation and cytokine production, via the paracrine interaction with its receptor RANK (*Bando et al., 2018*). On one side, we observed that the expression of RANK on ex vivo ILCPs was not detectable, whereas in vitro-expanded ILCPs acquire transient, intermediate levels of RANK after expansion (data not shown). Nevertheless, the contribution of RANKL to EC activation needs further investigation, as well as the formal evaluation of tm-TNF on ILCP surface. Of note, we observed higher levels of transcripts in in vitro-expanded ILCPs compared to their ex vivo counterparts, but very low levels of soluble TNF at the end of the expansion, suggesting that TNF might be present on the surface of expanded ILCPs. We might speculate that a sequential engagement of these ligand–receptor interactions occurs in the EC–ILCP interface, with initial tm-TNF/TNFR interactions that are needed to induce adhesion molecules expression, together with increased RANK expression in ECs. This could facilitate the sequential RANKL/RANK interactions, possibly required to support the production of pro-inflammatory cytokines, since the prevention of both TNFR/RANK engagement resulted in impaired cytokine production.

By performing a static adhesion assay, we show that the ILCP-mediated EC activation is functional. Therefore, ILCPs might favor the initial tethering of circulating immune cells to vascular ECs via E-Selectin induction, and the subsequent ICAM-1/LFA-1 and VCAM-1/VLA-4-mediated firm adhesion step, and support the EC-dependent recruitment of other immune cell types, thus facilitating their exit from the blood stream through the vessel wall.

As previously reported (*Eisenring et al., 2010*), NKp46+ ILCs were described to be crucial, in a subcutaneous melanoma mouse model, for the establishment of an IL-12-dependent anti-tumor immune response. A similar role was proposed for NKp44+ ILC3s in NSCLC patients (*Carrega et al., 2015*). Beside their putative role in supporting intratumoral TLS formation, an aspect that has been further recently supported in colorectal cancer patients (*Ikeda et al., 2020*), these cells were suggested to activate tumor-associated ECs and, in turn, favor leukocyte recruitment. Hence, ILCP–EC interactions might represent an early event during a large spectrum of biological reactions, ranging from inflammation, autoimmunity, and cancer. In tumors, leukocytes have to travel across the vessel

wall to infiltrate tumor tissue where they contribute to the killing of cancer cells. Further, the vessel wall serves as a barrier for metastatic tumor cells, and the integrity and the activation status of the endothelium serves as an important defense mechanism against metastasis formation (*Cedervall et al., 2015*).

The infiltration of immune cells in solid tumors often correlates with a better overall survival in cancer patients (*Zhang et al., 2003*, *Tjin and Luiten, 2014*, *Mina et al., 2015*). However, in the tumor microenvironment, ECs are dysfunctional and play a major role in several processes that contribute to cancer-associated mortality. One mechanism by which ECs can actively discourage the tumor homing of immune cells was described by Buckanovich and colleagues (*Buckanovich et al., 2008*). By transcriptionally profiling the tumor ECs (TECs) isolated from ovarian cancer specimens poorly infiltrated by T cells, the authors describe a mechanism that relies on the interaction between endothelin B receptor ($ET_B$R), found to be highly expressed by TECs, and its ligand endothelin-1 (ET-1), overexpressed in ovarian cancer cells. $ET_B$R signaling was shown to be responsible for the impaired ICAM-1-dependent T cell homing to tumors, and in turn, it correlated with shorter patient survival. Another mechanism that prevents T cell infiltration into the tumors relies on the overexpression of Fas ligand (FasL) on TEC surface (*Motz et al., 2014*) that causes the selective killing of tumor-specific $CD8^+$ T cells and to the accumulation of $FoxP3^+$ T regulatory (Tregs) cells within the tumors. Finally, it has been reported that Th1 cells can actively influence vessel normalization processes via the production of IFN-γ, which positively correlated with a more favorable outcome for cancer patients (*Tian et al., 2017*).

Therefore, committed ILCPs might represent an additional key regulator of efficient immune cell penetration into the tumor.

Tumors can engage multiple mechanisms to discourage the establishment of anti-tumor immune responses (*Vinay et al., 2015*). The shaping of an immunosuppressive milieu together with the diversion of the vascular system supports tumor progression and favors metastatic dissemination (*Hida and Maishi, 2018*). Here we show that RORγt-expressing ILCs, which share the transcription factor with in vitro-expanded ILCPs, infiltrate both human low-grade bladder and colon cancers and are associated with $CD31^+$ vessels, arguing for a potential ILC–EC interaction also in vivo. In vitro, we observed that the ability of ILCPs to induce adhesion molecules on ECs was dampened after the co-culture with bladder- and colon-derived tumor cells. ILCs are very plastic cells (*Bal et al., 2020*), and it has been reported that, in the cancer setting, tumor-derived TGF-β drives the transition of NK cells to dysfunctional and pro-tumoral ILC1s in vivo, a novel mechanism exploited by tumors to prevent the establishment of an innate anti-tumor response (*Gao et al., 2017*). One can speculate that a similar conversion also occurs for ILCPs toward a non-EC activating ILC subset. We showed that the mechanism of impairment of ILCPs might rely on adenosine. In vitro-expanded ILCPs express high mRNA levels of the adenosine receptors and CD39 at the protein level, whereas bladder cancer cells express CD73 and potentially also CD39. The presence of these two ectoenzymes, key for adenosine production, suggest that adenosine might be produced during the co-culture between cancer cells and ILCPs and impact ILCP functions. Indeed, by pre-exposing ILCPs to 2-Chloroadenosine, we could observe reduced EC-activating ability.

In conclusion, our data show that ILCPs, upon proper stimulation, might represent novel players in regulating the trafficking of immune cells to tissues, not only during the early phase of inflammation, but also at early phases of anti-tumor immune responses. Such contact-mediated events may be crucial in supporting further EC activation, to favor tumor-specific T-cell adhesion and, in turn, recruitment to the tumor site.

## Materials and methods

**Key resources table**

| Reagent type (species) or resource | Designation | Source or reference | Identifiers | Additional information |
|---|---|---|---|---|
| Cell line (*Homo sapiens*) | HUVEC (normal, adult, single donor) | Lonza | Cat# LZ-CC-2517 | Primary cell line |

*Continued on next page*

Continued

| Reagent type (species) or resource | Designation | Source or reference | Identifiers | Additional information |
|---|---|---|---|---|
| Cell line (*Homo sapiens*) | HDBEC (normal, adult, single donor) | Promocell. | Cat# C-12225 | Primary cell line |
| Cell line (*Homo sapiens*) | BU68.08 | This paper | | Primary cell line generated in L. Derré Lab from TaG2 stage cancer patient |
| Cell line (*Homo sapiens*) | TCC-Sup | | RRID:CVCL_1738 | Primary cell line Gift of G.N-Thalmann, Inselspital, Bern, Switzerland |
| Cell line (*Homo sapiens*) | SW1116 | ATCC | RRID:CVCL_0544 | |
| Biological sample (*Homo sapiens*) | Peripheral blood (adult, healthy donors) | Interregional Blood Transfusion SRC (Route de la Corniche 2, 1066 Epalinges) | | 9 mL Li Heparin tubes |
| Antibody | Alexa 488 anti-human CXCR3 (mouse monoclonal) | Biolegend | RRID:AB_10962442 | FACS/FC (1:50) |
| Antibody | FITC anti-human CD3 (mouse monoclonal) | Biolegend | RRID:AB_2562046 | FACS/FC (1:100) |
| Antibody | FITC anti-human CD4 (mouse monoclonal) | Biolegend | RRID:AB_2562052 | FACS/FC (1:100) |
| Antibody | FITC anti-human CD8 (mouse monoclonal) | Biolegend | RRID:AB_1877178 | FACS/FC (1:100) |
| Antibody | FITC anti-human CD14 (mouse monoclonal) | Biolegend | RRID:AB_2571929 | FACS/FC (1:100) |
| Antibody | FITC anti-human CD15 (mouse monoclonal) | Biolegend | RRID: AB_314196 | FACS/FC (1:100) |
| Antibody | FITC anti-human CD16 (mouse monoclonal) | Biolegend | RRID:AB_314206 | FACS/FC (1:100) |
| Antibody | FITC anti-human CD19 (mouse monoclonal) | Biolegend | RRID:AB_2750099 | FACS/FC (1:100) |
| Antibody | FITC anti-human CD20 (mouse monoclonal) | Biolegend | RRID:AB_314252 | FACS/FC (1:100) |
| Antibody | FITC anti-human CD31 (mouse monoclonal) | Biolegend | RRID:AB_314330 | FC (1:100) |
| Antibody | FITC anti-human CD33 (mouse monoclonal) | Biolegend | RRID:AB_314344 | FACS/FC (1:25) |
| Antibody | FITC anti-human CD34 (mouse monoclonal) | Biolegend | RRID:AB_1732005 | FACS/FC (1:50) |
| Antibody | FITC anti-human CD94 (mouse monoclonal) | Miltenyi | RRID:AB_2659623 | FACS/FC (1:25) |
| Antibody | FITC anti-human CD203c (mouse monoclonal) | Biolegend | RRID:AB_11218991 | FACS/FC (1:100) |
| Antibody | FITC anti-human FɛcRI (mouse monoclonal) | Biolegend | RRID:AB_1227653 | FACS/FC (1:50) |
| Antibody | PE anti-human BAFF (mouse monoclonal) | Biolegend | RRID:AB_830752 | FC (1:50) |
| Antibody | PE anti-human CD4 (mouse monoclonal) | Biolegend | RRID:AB_2562053 | FACS/FC (1:50) |

*Continued*

| Reagent type (species) or resource | Designation | Source or reference | Identifiers | Additional information |
|---|---|---|---|---|
| Antibody | PE anti-human CD62E (mouse monoclonal) | Biolegend | RRID:AB_536008 | FC (1:100) |
| Antibody | PE anti-human CRTH2 (mouse monoclonal) | Biolegend | RRID:AB_10900060 | FACS/FC (1:100) |
| Antibody | PE anti-human IDO-1 (mouse monoclonal) | Invitrogen | RRID:AB_2572712 | FC (1:50) Intracellular |
| Antibody | PE anti-human RANK (mouse monoclonal) | R and D | RRID:AB_10643566 | FC (1:100) |
| Antibody | PE anti-human RANKL (mouse monoclonal) | Biolegend | RRID:AB_2256265 | FC (1:50) |
| Antibody | PE anti-human RORγt (mouse monoclonal) | BD | RRID:AB_2686896 | FC (1:25) Intracellular |
| Antibody | PE-CF594 anti-human CD14 (mouse monoclonal) | BD | RRID:AB_11153663 | FC (1:400) |
| Antibody | PE-CF594 anti-human T-bet (mouse monoclonal) | BD | RRID:AB_2737621 | FC (1:25) Intracellular |
| Antibody | PE-Dazzle anti-human CD39 (mouse monoclonal) | Biolegend | RRID:AB_2564318 | FC (1:200) |
| Antibody | PerCP-Cy5.5 anti-human CCR4 (mouse monoclonal) | Biolegend | RRID:AB_2562391 | FACS/FC (1:100) |
| Antibody | PerCP-Cy5.5 anti-human CD28 | Biolegend | RRID:AB_2073718 | FC (1:100) |
| Antibody | PerCP-Cy5.5 anti-human NKp44 (mouse monoclonal) | Biolegend | RRID:AB_2616752 | FC (1:25) |
| Antibody | PE-Cy5 anti-human CD106 (mouse monoclonal) | Biolegend | RRID:AB_2214227 | FC (1:100) |
| Antibody | PE-Cy7 anti-human CCR6 (mouse monoclonal) | Biolegend | RRID:AB_10916518 | FACS/FC (1:100) |
| Antibody | PE-Cy7 anti-human CD4 (mouse monoclonal) | BC | Cat # 737660 Clone 7975048 | FC (1:400) |
| Antibody | PE-Cy7 anti-human CD62E (mouse monoclonal) | Biolegend | RRID:AB_2800891 | FC (1:50) |
| Antibody | PE-Cy7 anti-human NKp46 | BD | RRID:AB_10894195 | FC (1:50) |
| Antibody | APC anti-human CD3 (mouse monoclonal) | BC | RRID:AB_130788 | FC (1:100) |
| Antibody | APC anti-human CD30L (mouse monoclonal) | R and D | RRID:AB_416825 | FC (1:100) |
| Antibody | APC anti-human c-Kit (mouse monoclonal) | BD | RRID:AB_398461 | FACS/FC (1:50) |
| Antibody | APC anti-human GATA3 (mouse monoclonal) | Biolegend | RRID:AB_2562725 | FC (1:50) Intracellular |
| Antibody | Alexa Fluor 700 anti-human CD4 (mouse monoclonal) | Biolegend | RRID:AB_493743 | FACS/FC (1:400) |
| Antibody | Alexa Fluor 700 anti-human CD16 (mouse monoclonal) | Biolegend | RRID:AB_2278418 | FC (1:100) |

*Continued on next page*

*Continued*

| Reagent type (species) or resource | Designation | Source or reference | Identifiers | Additional information |
|---|---|---|---|---|
| Antibody | Alexa Fluor 700 anti-human CD45RA (mouse monoclonal) | BD | RRID:AB_1727496 | FACS/FC (1:100) |
| Antibody | APC-Cy7 anti-human CXCR5 (mouse monoclonal) | Biolegend | RRID:AB_2562593 | FC (1:100) |
| Antibody | APC-H7 anti-human CD19 (mouse monoclonal) | BD | RRID:AB_1645728 | FC (1:100) |
| Antibody | APC/Fire750 anti-human CD45RO (mouse monoclonal) | Biolegend | RRID:AB_2616717 | FACS/FC (1:100) |
| Antibody | eFluor450 anti-human CD73 (mouse monoclonal) | eBioscience | RRID:AB_11041811 | FC (1:200) |
| Antibody | Pacific Blue anti-human CD54 (mouse monoclonal) | Biolegend | RRID:AB_10900234 | FC (1:100) |
| Antibody | BV421 anti-human CXCR5 (mouse monoclonal) | Biolegend | RRID:AB_2562302 | FACS/ FC (1:100) |
| Antibody | BV421 anti-human CD127 (mouse monoclonal) | Biolegend | RRID:AB_10960140 | FACS/FC (1:100) |
| Antibody | BV421 anti-human NRP1 (mouse monoclonal) | Biolegend | RRID:AB_2562361 | FC (1:100) |
| Antibody | BV650 anti-human CCR6 (mouse monoclonal) | Biolegend | RRID:AB_2562235 | FC (1:100) |
| Antibody | BV650 anti-human CD62L (mouse monoclonal) | Biolegend | RRID:AB_2561461 | FC (1:100) |
| Antibody | BV650 anti-human CD69 (mouse monoclonal) | Biolegend | RRID:AB_2563158 | FC (1:100) |
| Antibody | BV711 anti-human CD40L (mouse monoclonal) | Biolegend | RRID:AB_2563845 | FC (1:100) |
| Antibody | Purified anti-LTα/β | Abcam | RRID:AB_2050404 | FC (1:25) |
| Antibody | Alexa 647 goat anti-mouse IgG (H+L) secondary antibody | Invitrogen | RRID:AB_2535804 | FC (1:800) |
| Antibody | Anti-human CD31 (mouse monoclonal) | Cell Marque | RRID:AB_629040 | IH (1:20) |
| Antibody | Anti-human CD3 (mouse monoclonal) | Ventana | Clone 2GV6 | IH (1:20) |
| Antibody | Anti-human RORγt (mouse monoclonal) | Millipore | RRID:AB_11205416 | IH (1:20) |
| Peptide, recombinant protein | rhIL-1β | PeproTech | Cat# 200-01B | |
| Peptide, recombinant protein | rhIL-2 | PeproTech | Cat# 200–02 | |
| Peptide, recombinant protein | rhIL-12 | PeproTech | Cat# 200–12 | |
| Peptide, recombinant protein | rhIL-21 | PeproTech | Cat# 200–21 | |
| Peptide, recombinant protein | rhTNF | PeproTech | Cat# 300-01A | |

*Continued on next page*

*Continued*

| Reagent type (species) or resource | Designation | Source or reference | Identifiers | Additional information |
|---|---|---|---|---|
| Peptide, recombinant protein | hrTNFR1:Fc | This paper | | Provided by PS |
| Peptide, recombinant protein | hrTNFR2:Fc | This paper | | Provided by PS |
| Peptide, recombinant protein | hrRANK:Fc | Adipogen | Cat# AG-40B-0018-C050 | |
| Chemical compound, drug | BAY 11–7082 | Adipogen | Cat# AG-CR1-0013-M010 | |
| Chemical compound, drug | 2-Chloroadenosine | Sigma | Cat# C5134 | |
| Sequenced-based reagent | hA2A_F | NCBI Nucleotide | PCR primers | CTCCGG TACAATGGC TTGGT |
| Sequenced-based reagent | hA2A_R | NCBI Nucleotide | PCR primers | TGGTTC TTGCCCTCC TTTGG |
| Sequenced-based reagent | hA2B_F | NCBI Nucleotide | PCR primers | A TGCCAACAGC TTGAATGGAT |
| Sequenced-based reagent | hA2B_R | NCBI Nucleotide | PCR primers | GAGGTCACC TTCCTGGCAAC |
| Sequenced-based reagent | hA3_F | NCBI Nucleotide | PCR primers | |
| | TTGACCAAAAGGAGGAGAAGT | Sequenced-based reagent | hA3_R | NCBI Nucleotide |
| PCR primers | AGTCACATCTGTTCAGTAGGAG | | | |
| Sequenced-based reagent | hIL-6_F | NCBI Nucleotide | PCR primers | GGATTCAA TGAGGAGAC TTGC |
| Sequenced-based reagent | hIL-6_R | NCBI Nucleotide | PCR primers | GTTGGG TCAGGGGTGG TTAT |
| Sequenced-based reagent | hIL-8_F | NCBI Nucleotide | PCR primers | AGCTCTGTG TGAAGG TGCAG |
| Sequenced-based reagent | hIL-8_R | NCBI Nucleotide | PCR primers | TGGGG TGGAAAGG TTTGGAG |
| Sequenced-based reagent | hGM-CSF_F | NCBI Nucleotide | PCR primers | GCCTCAGC TACG TTCAAGG |
| Sequenced-based reagent | hGM-CSF_R | NCBI Nucleotide | PCR primers | CATAGGAG TTAGG TCCCCACA |
| Sequenced-based reagent | hIFN-γ_F | NCBI Nucleotide | PCR primers | TGCCTTCCCTG TTTTAGCTGC |
| Sequenced-based reagent | hIFN-γ_R | NCBI Nucleotide | PCR primers | TCGGTAAC TGACTTGAATG TC |

*Continued on next page*

*Continued*

| Reagent type (species) or resource | Designation | Source or reference | Identifiers | Additional information |
|---|---|---|---|---|
| Sequenced-based reagent | hTNF_F | NCBI Nucleotide | PCR primers | |
| | GAGGCCAAGCCCTGGTATG | Sequenced-based reagent | hTNF_R | NCBI Nucleotide |
| PCR primers | CGGGCCGATTGATCTCAGC | | | |
| Commercial assay or kit | LIVE/DEAD Fixable Aqua Dead Cell Stain Kit | Thermo Fisher | Cat# L34957 | FC (1:500) |
| Commercial assay or kit | Foxp3 / Transcription Factor Staining Buffer Set | eBioscience | Cat# 00-5523-00 | |
| Commercial assay or kit | KAPA SYBR FAST qPCR KITs | KAPA Biosystems | Cat# 4385612 | |
| Software, algorithm | Eco Real-Time PCR System Software | Illumina | Cat# EC-101–1001 | |
| Software, algorithm | EcoStudy Software | Illumina | EcoStudy 5.0.4883 | |
| Software, algorithm | FlowJo | TreeStar | RRID:SCR_008520 | |
| Software, algorithm | Prism | GraphPad | RRID:SCR_002798 | |
| Other | CountBright Absolute Counting Beads | Thermo Fisher | Cat# C36950 | |
| Other | EGM Endothelial Growth Medium BulletKit | Lonza | Cat# LZ-CC-3124 | |

## Cell isolation

Human circulating ILCs and naïve CD4$^+$ T cells were isolated from PB mononuclear cells (PBMCs) of healthy donors by density-gradient centrifugation on 1.077 g/mL Ficoll-Hypaque (Lymphoprep) and ex vivo sorting. Individual human ILC subsets were isolated by fluorescence activated cell sorting (FACS) using the following antibodies: FITC anti-CD3 (Biolegend), -CD4 (Biolegend), -CD8 (Biolegend), -CD14 (Biolegend), -CD15 (Biolegend), -CD16 (Biolegend), -CD19 (Biolegend), -CD20 (Biolegend), -CD33 (Biolegend), -CD34 (Biolegend), -CD94 (Miltenyi), -CD203c (Biolegend) and -FcεRI (Biolegend) (lineage markers); PE anti-CRTH2 (Biolegend); APC anti-c-Kit (BD); and BV421 anti-CD127 (Biolegend). ILC subsets were sorted within the Lin$^-$ CD127$^+$ fraction, according to the expression of c-Kit and CRTH2: ILC1s as c-Kit$^-$CRTH2$^-$cells; ILC2s as c-Kit$^{+/-}$ CRTH2$^+$ cells and ILCPs as c-Kit$^+$ CRTH2$^-$ cells. Naïve total CD4$^+$ T cells were first isolated by FACS by using FITC anti-CD3 (Biolegend), PE anti-CD4 (Biolegend), and Alexa Fluor 700 anti-CD45RA antibodies (BD). Following in vitro expansion, individual CD4$^+$ Th cell subsets were isolated by FACS using Alexa 488 anti-CXCR3 (Biolegend), PerCP-Cy5.5 anti-CCR4 (Biolegend), PE anti-CRTH2 (Biolegend), PE-Cy7 anti-CCR6 (Biolegend), APC anti-CD3 (BC), Alexa Fluor 700 anti-CD4 (Biolegend), APC/Fire750 anti-CD45RO (Biolegend), and BV421 anti-CXCR5 (Biolegend) antibodies. Gating on CD3$^+$CD4$^+$-CD45RO$^+$CXCR5$^-$ cells, the Th subsets were sorted as follows: Th1 as CRTH2$^-$CXCR3$^+$CCR6$^-$ cells; Th* as CRTH2$^-$CXCR3$^+$CCR6$^+$ cells; Th2 as CRTH2$^+$CXCR3$^-$CCR6$^-$cells; Th17 as CRTH2$^-$CXCR3$^-$CCR4$^+$CCR6$^+$ cells. Individual ILC subsets, naïve CD4$^+$ T cells, and individual CD4$^+$ Th cell subsets were all isolated by FACS on a FACS Aria II or a FACS Aria III (BD).

## Cell culture and blocking experiments

Highly purified ILC subsets (≥90%) were expanded in vitro for at least 2 weeks in the presence of 100 U/mL of rh-IL-2 (PeproTech), 1 µg/mL of phytohaemagglutinin (PHA – PeproTech) and irradiated allogenic feeder cells obtained from three different donors (1:10 ILC/feeder cell ratio) in RPMI-1640 (Gibco) supplemented with 8% human serum (HS), 1% penicillin/streptomycin (10,000 U/mL, Gibco), 1% L-glutamine (Gibco), 1% nonessential amino acids (Gibco), 1% Na pyruvate (Gibco), 1%

Kanamycin 100× (Gibco), and 0.1% 2β-mercaptoethanol 500 mM (Sigma). After expansion, content of ILC subset in the cultures was assessed by flow cytometry and, if necessary, re-sorted to obtain pure (≥90%) ILC1s, ILC2s, and ILCPs, before being employed in co-culture experiments. Similarly, CD45RA$^+$ naïve CD4$^+$ T cells were first ex vivo isolated and in vitro-expanded for 2 weeks in the presence of 100 U/mL of rh-IL-2, 1 µg/mL of PHA, and irradiated allogenic feeder cells obtained from three different donors (1:10 CD4$^+$ T cell/feeder cell ratio) in RPMI-8% HS. Subsequently, individual CD4$^+$ Th cell subsets (i.e., Th1, Th2, Th17, and Th*) were isolated by FACS and cultured for additional 2 weeks in RPMI-8% HS in the presence of 100 U/mL of rh-IL-2 for Th1 and Th2, 20 U/mL of rh-IL-2 for Th17, 10 U/mL of rh-IL-2 with 50 ng/mL of rh-IL-12 and rh-IL-21 (Peprotech) for Th*. Primary human umbilical cord vein ECs (HUVECs – Lonza) and primary HDBECs (Promocell) were cultured in supplemented EC growth medium (EGM Ready To Use, Lonza) and used between passages 4 and 6. Non-muscle invasive bladder carcinoma cells (BU68.08) (EC number 2019–00564), muscle-invasive bladder carcinoma cells (TCC-Sup), and the colon adenocarcinoma cells (SW1116) were maintained in RPMI-1640 (Gibco) supplemented with 10% fetal calf serum (FCS), 1% penicillin/streptomycin (10,000 U/mL, Gibco), 1,15% AAG (Arg,Asp,Glu), 1% Hepes buffer 1M (Gibco), and 0.2 g/L ciproxin (Bayer). The used primary cell lines were checked for authenticity via STR profiling and periodically tested for mycoplasma contamination. Prior exposure to ILCPs, EC monolayers were incubated during 1 hr in the presence of 2.5 µM of BAY 11–7082 (Adipogen) in EC growth medium to specifically prevent NF-κB activation in ECs. EC monolayers where then washed once with PBS, before incubation with ILCPs at 1:1 ratio. Similarly, blocking experiments with the use of soluble Fc fusion proteins were performed: when indicated, ILCPs were pre-incubated for 30 min with 2 µg/mL of TNFR1:Fc, 5 µg/mL of TNFR2:Fc, 5 µg/mL of RANK:Fc (Adipogen), either alone or in combination. Then, ILCPs were washed once with PBS, then added to EC monolayers. Finally, when indicated, in vitro-expanded ILCPs were incubated overnight with 50 µM of 2-chloroadenosine (CADO, Sigma) in RPMI-8% HS with 10 U/mL of IL-2 prior co-culture with ECs.

## Co-culture experiments

Following expansion, individual pure (≥90%) ILC and Th subsets were rested overnight in RPMI-8% HS medium supplemented with 10 U/mL of rh-IL-2. Then, confluent EC monolayers were either co-cultured for 3 hr with individual ILC and Th subsets at 1:1 ratio, treated with 20 ng/mL of rh-TNF (Peprotech) or left untreated as positive and negative controls, respectively. Co-cultures of ECs with ILCPs were performed both in the presence or absence of 0.4 µm pore polycarbonate filter in 24-well transwell chambers (Corning). ILCPs were also incubated overnight with epithelial bladder carcinoma cell lines in RPMI 8% HS with 10 U/mL of IL-2 at 1:1 ratio, before exposure to EC monolayers. The day of the experiment, ILCs were collected, washed with PBS, and re-suspended in the respective EC growth medium (Lonza). At least three independent experiments were performed, using individual ILC and Th subsets isolated from a different donor. At the end of the experiment, supernatants were collected and stored at −20°C, and ECs were washed twice with PBS and detached with Accutase (Gibco) for 5 min at 37°C. Cell suspensions were then washed with PBS and stained for flow cytometry analyses.

## Phenotypic characterization

The phenotypic characterization of both ex vivo- and in vitro-expanded ILCPs from HDs, as well as the quantification of ex vivo ILCPs in the PB of bladder cancer patients, was performed by using the same antibodies as the ones used for isolation by FACS together with the following antibodies: PE anti-BAFF (Biolegend), -RANKL (Biolegend) and –RORγt (BD); PE-CF594 anti-T-bet (BD); PE-Dazzle anti-CD39 (Biolegend); PerCP-Cy5.5 anti-CD28 (Biolegend) and anti-NKp44 (Biolegend); PE-Cy7 anti-NKp46 (BD); APC anti-CD30L (R and D) and anti-GATA3 (Biolegend); Alexa Fluor 700 anti-CD45RA (Biolegend); Biotin anti-LTα$_1$β$_2$ (Abcam); APC/Fire750 anti-CD45RO (Biolegend) and APC-Cy7 anti-CXCR5 (Biolegend); eFluor450 anti-CD73 (eBioscience); BV421 anti-NRP1 (Biolegend); BV650 anti-CD62L (Biolegend), -CD69 (Biolegend) and anti-CCR6 (Biolegend); BV711 anti-CD40L (Biolegend). The activation state of ECs was assessed by flow cytometry using FITC anti-CD31 (Biolegend), PE anti-RANK (R and D), PE-Cy7 anti-CD62E (or E-Selectin – Biolegend), Pacific Blue anti-CD54 (or ICAM-1 – Biolegend) and PE-Cy5 anti-CD106 (or VCAM-1 – Biolegend). For the static adhesion assay, the assessment of PBMCs adhesion to ECs and the EC activation state was analyzed

using the following panel of antibodies: APC anti-CD3 (BC), PE-Cy7 anti-CD4 (BC), PE-CF594 anti-CD14 (BD), Alexa Fluor 700 anti-CD16 (Biolegend), APC-H7 anti CD19 (BD), FITC anti-CD31 (Biolegend), Pacific Blue anti-CD54 (Biolegend), PE anti-CD62E (Biolegend), and PE-Cy5 anti-CD106 (Biolegend). For the characterization of tumor cells, the following antibodies were used: PE-Dazzle anti-CD39 (Biolegend); eFluor450 anti-CD73 (eBioscience), and PE anti-IDO-1 (Invitrogen). All analyses included size exclusion (forward scatter [FSC] area versus side scatter [SSC] area), doublets exclusion (FSC height/ FSC area), and dead cell exclusion (LIVE/DEAD Fixable Aqua Dead Cell Stain Kit, Thermo Fisher). A minimum of 10,000 events were acquired on either a Gallios Cytometer (Beckman Coulter) or SORPLSR-II Cytometer (BD) and analyzed with FlowJo software (TreeStar).

## Static adhesion assay

ECs were plated at 80% confluency in a 24-well plate in complete EGM. Once adherent, the media was removed, ECs were washed with PBS, and 500 µL of complete EGM, containing or not ILCPs at 1:1 ratio, were added to the wells during 3 hr. As positive control, ECs were incubated during 3 hr with 20 ng/mL of TNF. After the co-culture, ECs were detached and stained with FITC anti-CD31 antibody and FACS-sorted to remove adherent ILCPs. Recovered ECs were seeded in a 48-well plate and let to adhere overnight in complete EGM. The morning after, the static adhesion assay was performed (adapted from *Safuan et al., 2012*). The adhesion of freshly isolated PBMCs was assessed by adding 4:1 cells (PBMC:EC) /well for 30 min at 37°C. Non-adherent cells were washed away from the EC monolayer by performing 2× washing steps with PBS. ECs, together with adherent PBMCs, were detached with Accutase (Gibco), and stained for flow cytometry analyses. The number of CD3, CD4, CD8, CD14, $CD56^{dim}CD16^+$, $CD56^{bright}CD16^{low}$, and CD19 expressing cells, as well as $Lin^-CD127^+$ total ILCs were quantified by flow cytometry by adding 10 µL of CountBright Absolute Counting Beads (Thermo Fisher) to the cell suspensions. 2000 beads/sample were acquired and cell counts normalized.

## RNA purification and qPCR

Total RNA was isolated from highly pure ex vivo- and in vitro-expanded ILCPs, from primary ECs (HUVECs) and from sorted human ILC and CD4 Th cell subsets using the TRIZOL reagent according to the manufacturer's instructions (Invitrogen, Carlsbad, CA, USA). Final preparation of RNA was considered protein-free if the ratio of spectrophotometer (NanoDrop, ThermoFischer, Carls- bad, CA, USA) readings at 260/280 nm was ≥1.7. Isolated mRNA was reverse-transcribed using the iScript cDNA Synthesis Kit (Bio-Rad Laboratories, Watford, UK) according to the manufacturer's protocol. The qPCR was carried out in the ECO Real-time PCR System (Illumina) with specific primers (see Key Resources Table) using KAPA SYBR FAST qPCR Kits (KAPA Biosystems, Inc, MA). Samples were amplified simultaneously in triplicate in one-assay run with a nontemplate control blank for each primer pair to control for contamination or for primer dimerization, and the Ct value for each experimental group was determined. The housekeeping gene (ribosomal protein S16) was used as an internal control to normalize the Ct values, using the $2^{-\Delta Ct}$ formula.

## Immunohistochemical staining

Immunohistochemical staining was performed on 2 µm paraffin sections with an automated IHC staining system (Ventana BenchMark ULTRA, Ventana Medical Systems, Italy). Sequential double IHC was performed on Ventana BenchMark ULTRA, using a ultraView Universal DAB detection Kit as the first stain and ultraView Universal Alkaline phosphatase Red detection kit as the second stain. Heat-induced epitope retrieval pre-treatment was performed using CC1 buffer (standard CC1, Roche Ventana) by boiling for 36 min for both CD31 and CD3 and for 64 min for RORγt. Afterwards, slides were incubated with primary antibodies: CD31 antibody (clone JC70, Cell Marque, dilution 1:20) for 16 min at 37°C or CD3 (clone 2GV6, Ventana, dilution 1:20) for 44 min at 37°C and RORγt (clone 6F3.1, Millipore, dilution 1:20) for 36 min at 37°C. CD31 and CD3 were visualized with DAB chromogen, and RORγt was visualized with Fast Red chromogen.

## Statistical analyses

GraphPad Prism 7 software was used to perform the statistical analyses. Paired or unpaired t-tests were used when comparing two groups. ANOVAs or the non-parametric Kruskal–Wallis test were

used for comparison of multiple groups. Data in graphs represent the mean ± SEM, with a p-value <0.05 (two-tailed) being significant and labeled with *. p-values <0.01, <0.001, or <0.0001 are indicated as **, ***, and ****, respectively. Without mention, differences are not statistically significant.

## Acknowledgements

This work was supported by grants from the Swiss National Science Foundation (PRIMA PR00P3_179727) to CJ; the Swiss Cancer League (KFS-4402-02-2018) to CJ; from a Bourse Pro-Femmes, University of Lausanne to ST. PS was supported by the Swiss National Science Foundation (310030A_176256); from Compagnia di San Paolo (2019.866) to EM and SC and from Fondazione Associazione Italiana per la Ricerca sul Cancro (AIRC 5 × 1000-21147 and AIRC IG) to EM and SC. PR was supported in part by grants from the Swiss National Science Foundation (FNS 31003A_156469 and FNS 310030_182735).

## Additional information

### Funding

| Funder | Grant reference number | Author |
| --- | --- | --- |
| Schweizerischer Nationalfonds zur Förderung der Wissenschaftlichen Forschung | PRIMA PR00P3_179727 | Camilla Jandus |
| Schweizerischer Nationalfonds zur Förderung der Wissenschaftlichen Forschung | FNS 31003A_156469 | Pedro Romero |
| Schweizerischer Nationalfonds zur Förderung der Wissenschaftlichen Forschung | 310030A_176256 | Pascal Schneider |
| Compagnia di San Paolo | 2019.866 | Simona Candiani Emanuela Marcenaro |
| Associazione Italiana per la Ricerca sul Cancro | AIRC 5x1000-21147 | Simona Candiani Emanuela Marcenaro |

The funders had no role in study design, data collection and interpretation, or the decision to submit the work for publication.

### Author contributions

Giulia Vanoni, Giuseppe Ercolano, Conceptualization, Data curation, Formal analysis, Validation, Investigation, Visualization, Methodology, Writing - original draft, Project administration, Writing - review and editing; Simona Candiani, Mariangela Rutigliani, Emanuela Marcenaro, Resources, Data curation, Formal analysis, Investigation, Writing - review and editing; Mariangela Lanata, Investigation, Methodology, Writing - review and editing; Laurent Derré, Resources, Formal analysis, Funding acquisition, Validation, Investigation, Methodology, Writing - review and editing; Pascal Schneider, Conceptualization, Resources, Funding acquisition, Methodology, Writing - review and editing; Pedro Romero, Conceptualization, Resources, Supervision, Funding acquisition, Methodology, Writing - review and editing; Camilla Jandus, Conceptualization, Supervision, Funding acquisition, Visualization, Project administration, Writing - review and editing; Sara Trabanelli, Conceptualization, Supervision, Funding acquisition, Validation, Visualization, Project administration, Writing - review and editing

### Author ORCIDs

Giulia Vanoni https://orcid.org/0000-0003-3199-2412
Sara Trabanelli https://orcid.org/0000-0001-8648-1324

### Decision letter and Author response

Decision letter https://doi.org/10.7554/eLife.58838.sa1

Author response https://doi.org/10.7554/eLife.58838.sa2

## Additional files

### Supplementary files

• Transparent reporting form

### Data availability

All data generated or analysed during this study are included in the manuscript and supporting files.

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
