## [Decision Letter]

**Acceptance summary:**

The work presented by Vanoni et al. presents novel and interesting results on ILC interaction with the endothelium favouring immune cell adhesion. The work is relevant and might have important clinical consequences.

**Decision letter after peer review:**

Thank you for submitting your article "Human primed ILCPs support endothelial activation through NF-κB signaling" for consideration by *eLife*. Your article has been reviewed by two peer reviewers, and the evaluation has been overseen by a Reviewing Editor and Tadatsugu Taniguchi as the Senior Editor. The reviewers have opted to remain anonymous.

The reviewers have discussed the reviews with one another and the Reviewing Editor has drafted this decision to help you prepare a revised submission.

The work presented by Vanoni et al. presents novel and interesting results on ILC interaction with the endothelium favoring immune cell adhesion. The manuscript is clearly written, easy to read and the conclusions are supported by the data depicted. However it raises some concerns and major points to be addressed. In fact, this work is done exclusively in vitro using cell lines and it is unclear what the real in vivo relevance is, it has unfortunately "the taste of truncated story". Some suggestions below might help to improve the manuscript and its narrative:

Essential revisions:

1) It is unclear under which circumstances the types of activated ILCP, that the authors describe after ex vivo expansions, actually appear in vivo. Can they e.g. be found in vasculature of inflamed/tumor tissues in vivo in humans?

2) Overall, the authors need to confirm at least some of their key findings using primary EC and/or cancer cells.

3) The definition of ILCP is naivety, i.e. non-differentiated phenotype and function. The authors show that ex vivo expanded ILCP differentiate and a proportion express RORgt indicative of ILC3 differentiation. The authors should also explore ILC1 and ILC2 differentiation in these cells. A more thorough characterization of ex vivo expanded ILCP is warranted. Depending on the outcome of that characterization, a more appropriate term instead of ILCP should be used downstream.

4) NFkB inhibition can have effects on cell viability, please show viability effects on ECs following NFkB inhibition

5) To fully understand the crosstalk between ILCs and EC the cellular source of cytokines presented in Supplementary Figure 3 would need to be unraveled. Intracellular cytokine stainings could be used to address this issue.

6) Figure 3E is based on two experiments and should be solidified by at least another 1-2 experiments.

7) In the PBMC-adhesion experiments, it is unclear what the mechanisms are. Is the adhesion also suppressed by inhibition of TNF/NFkB signaling in EC? This would enhance the value of these observations.

8) Similarly, the last pieces of data related to the bladder cancer cell lines are underdeveloped. To fully understand the ILC-EC crosstalk in the context of cancer, the authors should unravel the mechanism underlying the inhibition of ILC-induced adhesion molecule expression on EC in the presence of tumor cells. It is unclear how the co-culture experiment was performed and if the tumor cells have effects on ILCP and/or EC and what the mechanisms are.

---

## [Author Response]

Essential revisions:1) It is unclear under which circumstances the types of activated ILCP, that the authors describe after ex vivo expansions, actually appear in vivo. Can they e.g. be found in vasculature of inflamed/tumor tissues in vivo in humans?

This is a very interesting point. To address this question, we performed IHC staining on tumor tissue sections from low-grade transitional bladder carcinoma (revised Figure 6A, panels 1-4) and from low-grade invasive colon adenocarcinoma (new Figure 6—figure supplement 1A). As shown in both figures, RORγt^+^CD3^-^ ILCs (i.e., type 3 ILCs, the tissue counterpart of in vitro-expanded ILCPs that are also characterized by the expression of RORγt) are present in both tumors in close association with CD31^+^ vascular endothelial cells. Their presence was barely detected in high-grade bladder cancer sections (revised Figure 6A, panels 5-8) correlating with the more accentuated impairment of in vitro*-*expanded ILCPs to activate ECs when exposed to MIBC cells (revised Figure 6C). The main manuscript text has been edited in the Results section.

2) Overall, the authors need to confirm at least some of their key findings using primary EC and/or cancer cells.

Regarding the use of primary ECs, we would like to point out that HUVECs are in fact primary cells isolated from the human umbilical cord vein (Lidington et al., 1999). They are not immortalized and the window of time in which they can be used is very limited (between passages 4 and 6). To further substantiate our findings with additional endothelial cell lines, we performed similar experiments by employing another primary blood endothelial cell line, e.g., HDBECs (Human Dermal Blood Endothelial Cells – Promocell). We could confirm that in vitro-expanded ILCPs have EC activating capacities also in this setting, as shown in revised Figure 1—figure supplement 1B.

Regarding tumor cells: BU68.08 is a non-invasive bladder carcinoma cell line isolated from a patient with a TaG2 tumor (early stage) by our collaborator LD at the Department of Urology (University Hospital of Lausanne – CHUV). TCC-Sup is an invasive urothelium carcinoma cell line corresponding to G4 histological grade (late stage), kindly provided to LD by G.N. Thalmann, Inselspital, Bern, Switzerland.

Since we also observed infiltration of tumor tissue by RORγt^+^CD3^-^ ILCs in low-grade invasive colon adenocarcinoma patients (new Figure 6—figure supplement 1A), we decided to repeat the same experiments done with the patient-derived bladder cancer cells with the SW1116 colon cancer cell line. As shown in the new Figure 6—figure supplement 1B, the overnight incubation of in vitro-expanded ILCPs with SW1116 cancer cells reduced the ability of ILCPs to upregulate E-Selectin and VCAM-1 in ECs, although the reduction did not result to be significant. The main manuscript text has been edited in the Results section.

3) The definition of ILCP is naivety, i.e. non-differentiated phenotype and function. The authors show that ex vivo expanded ILCP differentiate and a proportion express RORgt indicative of ILC3 differentiation. The authors should also explore ILC1 and ILC2 differentiation in these cells. A more thorough characterization of ex vivo expanded ILCP is warranted. Depending on the outcome of that characterization, a more appropriate term instead of ILCP should be used downstream.

As shown in the revised Figure 1—figure supplement 1E, we analyzed the expression of T-bet and GATA3 in RORγt^+^ and RORγt^-^ ILCPs at the end of the in vitro expansion. No significant difference was observed between the two subpopulations. The observed expression of T-bet in approximately 20% and GATA3 in 40% of the cells suggests that they might maintain features of progenitor/multipotent cells. Moreover, we set up a flow cytometry panel to further characterize the phenotype of in vitro-expanded ILCPs, compared to their ex vivo counterparts. As shown in revised Figure 1—figure supplement 1C-D, in vitro-expanded ILCPs acquire the expression of CCR6 and upregulate CXCR5 levels, compared to their ex vivo counterparts. CCR6 is a chemokine receptor that binds to CCL20, and cells expressing it home to inflamed/infected tissues (Schutyser et al., 2003). In mice, CCR6 is a marker expressed on a subset of ILC3 which shares some features with fetal LTi cells (Melo-Gonzalez and Hepworth, 2017). In humans, CCR6 is expressed by a subset of Lin^-^ c-Kit^+^ ILC3s within cryptopatches (Lügering et al., 2010). CXCR5 is also a chemokine receptor expressed by LTi cells (Kim et al., 2016) and it was shown to be upregulated in circulating human ILC3s upon infection with *Mycobacterium tuberculosis*, together with increased plasma levels of its ligand CXCL13 (Ardain et al., 2019). As previously reported, Neuropilin1 (NRP1), that is a receptor for VEGFA and that is expressed by ILC3s with LTi features within lymphoid tissues, is not expressed by circulating ILC3s (Shikhagaie et al., 2017) and is detected neither in ex vivo nor in in vitro*-*expanded ILCPs. Regarding ILC biology, since CCR6 and CXCR5 expression is mostly found in LTi cells, these data suggest that in vitro-expanded ILCPs isolated from the PB of healthy donors show lymphoid-tissue inducer features. Compared to ex vivo, in vitro*-*expanded ILCPs downregulated CD28 expression. Finally, following stimulation with IL-1β and IL-23, we observed that in vitro*-*expanded ILCPs did produce IL-22, but not IL-17 (therefore being closer to the tissue ILC3 NCR^+^ subset). They could also produce IL-5 and IL-13, i.e., type2 cytokines, as well as IL-6 and IFN-γ (Author response image 1). Overall, despite the fact that in vitro-expanded ILCPs acquire to some extent LTi-like properties and are partially skewed towards an ILC3-like phenotype, the expression of T-bet and GATA-3 and the production of a variety of cytokines upon stimulation suggest that these cells preserve progenitor/multipotent cell features. Therefore, we believe that in vitro-expanded ILCPs is an appropriate term to refer to these cells.

**Author response image 1. respfig1:** Ex vivo and in vitro-expanded ILCPs were plated at 100cell/10μL density in RPMI-8%HS with 100U/mL of IL-2 and stimulated for 48h with 20ng/mL of IL-1β and IL-23. The cell-free supernatant was collected and analyzed for its cytokine content with the LEGENDplex technology (Biolegend) (n=3).

4) NFkB inhibition can have effects on cell viability, please show viability effects on ECs following NFkB inhibition.

As shown in the graphs in Author response image 2, the treatment with the NF-κB inhibitor BAY11-7082 did not affect EC viability at the concentration used to perform the experiments (i.e., 2.5 μΜ). At this concentration, a strong reduction of adhesion molecules expression in TNF-treated HUVECs could be achieved while maintaining cell viability (Author response image 2).

**Author response image 2. respfig2:** Effects of the NF-kB inhibitor BAY11-7082 on HUVEC cell viability and adhesion molecule expression. (A) HUVEC cell monolayers were treated during 1 h with different concentrations of the NF-κB inhibitor BAY11-7082 in complete EGM, or left untreated. First, the viability of ECs following treatment with NF-κB inhibitor was assessed by flow cytometry with LIVE/DEAD staining. (B) Following NF-κB inhibition, EC monolayers were treated with 20 ng/mL of TNF during 3 h. Then, ECs were harvested and analyzed for cell-surface adhesion molecule expression by flow cytometry. The graph shows the induction of the indicated adhesion molecules on the endothelial cell surface.

5) To fully understand the crosstalk between ILCs and EC the cellular source of cytokines presented in Supplementary Figure 3 would need to be unraveled. Intracellular cytokine stainings could be used to address this issue.

This is another very well taken point. To have more consistent data, we determined the source of cytokines that accumulate in the cell-free supernatant during the co-culture by performing qPCR on sorted cells post co-culture, since both intracellular staining and western blot analysis required prohibitively high cell numbers.

Therefore, after the 3 h co-culture, we sorted both CD31^+^ ECs and CD31^-^ ILCPs, and performed RNA extraction, RT and qPCR analyses. As controls, ILCPs at the end of the expansion and untreated ECs were used. As shown in revised Figure 2B, before co-culturing ECs and ILCPs, high levels of IFN-γ, GM-CSF and of TNF were only detected in ILCPs. After 3h co-culture, IL-6 and IL-8 were found to be specifically produced by ILCP-exposed ECs, whereas both cell types could produce IFN-γ and GM-CSF after co-culture.

6) Figure 3E is based on two experiments and should be solidified by at least another 1-2 experiments.

The experiment depicted in the original Figure 3E was repeated to have n=4. We have replaced Figure 3E with an updated version in the revised Figure 3E.

7) In the PBMC-adhesion experiments, it is unclear what the mechanisms are. Is the adhesion also suppressed by inhibition of TNF/NFkB signaling in EC? This would enhance the value of these observations.

As suggested, we performed the static adhesion assay on untreated/TNF-treated endothelial cells after blocking NF-κB pathway activation during 1 h prior to addition of the total PBMCs at 4:1 ratio. As shown in revised Figure 5D, the blockade of NF-κB activation before adding the PBMCs influenced the adhesion of all cell types analyzed (red empty square dots), although not significantly. Interestingly, when blocking NF-κB before treating ECs with TNF, adhesion of total PBMCs was completely impaired (red half-empty square dots). However, as explained in the Results section, NF-κB is key in upregulating adhesion molecules on EC surface that are necessary to facilitate adhesion of PBMCs. Therefore, we believe that since ECs could not upregulate adhesion molecules on their cell surface, this led to a dramatic reduction in adhesive events.

Taken together, these results suggest that the adhesion of PBMCs to ECs in vitro relies at least in part on NF-κB activation.

8) Similarly, the last pieces of data related to the bladder cancer cell lines are underdeveloped. To fully understand the ILC-EC crosstalk in the context of cancer, the authors should unravel the mechanism underlying the inhibition of ILC-induced adhesion molecule expression on EC in the presence of tumor cells. It is unclear how the co-culture experiment was performed and if the tumor cells have effects on ILCP and/or EC and what the mechanisms are.

We thank the reviewer for raising this point. We apologize if the set-up of the co-culture experiments was not clear. In detail, first, we mixed in vitro-expanded ILCPs with bladder cancer cells at 1:1 ratio during an overnight incubation, then ILCPs were collected and incubated with ECs at 1:1 ratio for 3 hours. Therefore, pre-exposure to bladder cancer cells affected ILCPs by impairing their EC activating capacities.

In order to unravel which could be the mechanisms behind the bladder cancer cell-mediated impairment of ILCPs, we considered several molecules known to exert immunosuppressive functions in the tumor microenvironment. In particular, adenosine and kynurenines are known to be strong immunosuppressive mediators in cancer (Pitt et al., 2016; Antonioli et al., 2013). Therefore, we analyzed the bladder cancer cells for the expression of CD39 and CD73 (two ectoenzymes involved in the conversion of ATP into AMP, and of AMP into adenosine, respectively) and IDO-1 (i.e., indoleamine 2,3-dioxygenase 1, enzyme involved in the tryptophan to kynurenine conversion). We observed CD73 expression in TCC-Sup MIBC cells that was even higher after overnight co-culture with ILCPs (revised Figure 6E). Interestingly, CD39 expression became detectable in TCC-Sup cells only after the overnight incubation with ILCPs (revised Figure 6E). Similarly, we observed CD39 expression in ILCPs at the end of the expansion that was also maintained after overnight exposure to TCC-Sup cells, while CD73 was not detectable on these cells (Figure 6—figure supplement 1C). Regarding IDO-1, its expression could not be detected in bladder cancer cells (revised Figure 6E). Therefore, the presence of CD39 and CD73 on both cell types suggests that adenosine might be produced and present in the supernatant during the overnight incubation between ILCPs and TCC-Sup cells, as a result of a concerted effort between tumor cells and ILCPs, as we previously described for CD39^+^Tregs and CD73^+^Teff in breast cancer (Gourdin et al., 2018). Despite adenosine is not measurable in the supernatant due to its very short half-life (Moser et al., 1989), to address if adenosine could affect ILCPs, we performed qPCR analysis for the adenosine receptors 2A (A2A), A2B and A3. Compared to ex vivo ILCPs, in vitro-expanded ILCPs upregulated the expression of all the three adenosine receptors, and in particular the expression of A2B (Figure 6—figure supplement 1D). Therefore, ILCPs express the receptors that bind adenosine and could be impacted in their function by adenosine. In line with this hypothesis, pre-exposure of ILCPs to 2-Chloroadenosine (CADO, a stabilized form of adenosine) reduced the ability of ILCPs to activate ECs (revised Figure 6F). Overall, these data suggest that tumor cells might impair EC-activating capacity of ILCPs, at least in part, via adenosine. The main manuscript text has been edited in the Results section.

References:

Antonioli L, Pacher P, Vizi ES, Hasko G. CD39 and CD73 in immunity and inflammation. Trends Mol Med. 2013;19(6):355–67.

Gourdin N, Bossennec M, Rodriguez C, Vigano S, Machon C, Jandus C, Bauché D, Faget J, Durand I, Chopin N, Tredan O, Marie JC, Dubois B, Guitton J, Romero P, Caux C, Ménétrier-Caux C. Autocrine Adenosine Regulates Tumor Polyfunctional CD73^+^CD4^+^ Effector T Cells Devoid of Immune Checkpoints. Cancer Res. 2018;78(13):3604-3618. doi: 10.1158/0008-5472.CAN-17-2405.

Kim, C.H., S. Hashimoto-hill, and M. Kim. 2016. Migration and Tissue Tropism of Innate Lymphoid Cells. Trends Immunol. 2016;37:68–79. doi:10.1016/j.it.2015.11.003.

Lidington EA, Moyes DL, McCormack AM, Rose ML. A comparison of primary endothelial cells and endothelial cell lines for studies of immune interactions. Transplant Immunology. 1999;7(4)239-246.

Lügering, A., M. Ross, M. Sieker, J. Heidemann, I.R. Williams, W. Domschke, and T. Kucharzik. CCR 6 identifies lymphoid tissue inducer cells within cryptopatches. Clin. Exp. Immunol. 2010;160:440–449. doi:10.1111/j.1365-2249.2010.04103.x.

Melo-Gonzalez, F., and M.R. Hepworth. Functional and phenotypic heterogeneity of group 3 innate lymphoid cells. Immunology. 2017;150:265–275. doi:10.1111/imm.12697.

Möser GH, Schrader J, Deussen A. Turnover of adenosine in plasma of human and dog blood. Am J Physiol 1989;256:C799-806. doi: 10.1152/ajpcell.1989.256.4.C799. Pitt JM, Vetizou M, Daillere R, et al. Resistance mechanisms to immune-checkpoint blockade in cancer: tumor-intrinsic and -extrinsic factors. Immunity. 2016;44(6):1255–69.

Shikhagaie, M.M., Å.K. Björklund, J. Mjösberg, J.S. Erjefält, A.S. Cornelissen, X.R. Ros, S.M. Bal, J.J. Koning, R.E. Mebius, M. Mori, M. Bruchard, B. Blom, and H. Spits. Neuropilin-1 Is Expressed on Lymphoid Tissue Residing LTi-like Group 3 Innate Lymphoid Cells and Associated with Ectopic Lymphoid Aggregates. Cell Rep. 2017;18:1761–1773. doi:10.1016/j.celrep.2017.01.063.

Ardain, A., R. Domingo-gonzalez, S. Das, W. Samuel, N.C. Howard, A. Singh, M. Ahmed, S. Nhamoyebonde, J. Rangel-Moreno, P. Ogongo, L. Lu, D. Ramsuran, M.D.L. Garcia-hernandez, T.K. Ulland, M. Darby, E. Park, F. Karim, L. Melocchi, R. Madansein, K.J. Dullabh, M. Dunlap, N. Marin-agudelo, T. Ebihara, and T. Ndung. Group 3 innate lymphoid cells mediate early protective immunity against tuberculosis. Nature 2019;570:528–532. doi:10.1038/s41586-019-1276-2.